# Similar neural responses predict friendship

Carolyn Parkinson [1], Adam M. Kleinbaum [2] & Thalia Wheatley[3]

Human social networks are overwhelmingly homophilous: individuals tend to befriend others who are similar to them in terms of a range of physical attributes (e.g., age, gender). Do similarities among friends reflect deeper similarities in how we perceive, interpret, and respond to the world? To test whether friendship, and more generally, social network proximity, is associated with increased similarity of real-time mental responding, we used functional magnetic resonance imaging to scan subjects' brains during free viewing of naturalistic movies. Here we show evidence for neural homophily: neural responses when viewing audiovisual movies are exceptionally similar among friends, and that similarity decreases with increasing distance in a real-world social network. These results suggest that we are exceptionally similar to our friends in how we perceive and respond to the world around us, which has implications for interpersonal influence and attraction.

[1] Department of Psychology, University of California, Los Angeles, 1285 Franz Hall, Box 951563 , Los Angeles, CA 90095, USA. [2] Tuck School of Business, Dartmouth College, 100 Tuck Hall, Hanover, NH 03755, USA. [3] Department of Psychological and Brain Sciences, Dartmouth College, 6207 Moore Hall, Hanover, NH 03755, USA. Correspondence and requests for materials should be addressed to C.P. (email: cparkinson@ucla.edu)

The notion that people tend to resemble their friends is an enduring intuition, as evidenced by the centuries-old adage, "birds of a feather flock together"[1]. Research has borne out this intuition: social ties are forged at a higher-than-expected rate between individuals of the same age, gender, ethnicity, and other demographic categories[2]. This assortativity in friendship networks is referred to as homophily and has been demonstrated across diverse contexts and geographic locations, including online social networks[2–5]. Indeed, consistent evidence suggests that homophily is an ancient organizing principle and perhaps the most robust empirical regularity of human sociality. Despite pressures to divide labor and otherwise organize complementary needs and roles in the kinds of social groups in which humans evolved, social ties in small hunter-gatherer bands reflect similarities, rather than differences, across a range of attributes, including age, weight, body fat, handgrip strength, and cooperative behavioral tendencies[4]. Significant examples of heterophily, which refers to the tendency to associate with others who are dissimilar from oneself, are markedly rarer in such groups. Consistent with its ancient history, homophily also characterizes the social networks of our close primate relatives[6] and has been suggested to confer advantages for cohesion, collective action, and empathy[4,6]. When humans do forge ties with individuals who are dissimilar from themselves, these relationships tend to be instrumental, task-oriented (e.g., professional collaborations involving people with complementary skill sets[7]), and short-lived, often dissolving after the individuals involved have achieved their shared goal[8]. Thus, human social networks tend to be overwhelmingly homophilous[8].

Despite robust evidence that homophily organizes human social networks, significant lacunae remain in our understanding of how homophily arises and functions in these networks[3,6]. Prior studies of homophily have been concerned largely with physical traits and demographic variables, such as age, gender, and class. Importantly, additional research has demonstrated that homophily extends beyond overt, demographic cues, to at least some aspects of behavior and personality. For example, behavioral tendencies (e.g., donations in public goods games) associated with altruistic behavior are more similar among individuals who are friends compared with those who are not[4], consistent with suggestions from evolutionary game theory that altruistic behavior only benefits individuals if their interaction partners also behave altruistically[9,10]. Remarkably, social network proximity is as important as genetic relatedness and more important than geographic proximity in predicting the similarity of two individuals' cooperative behavioral tendencies[4]. Thus, although prior research on homophily focused largely on relatively coarse variables, such as demographic categories, a growing body of evidence has begun to move beyond externally evident demographic attributes, and suggests that social network proximity can be a powerful predictor of behavioral similarity.

In addition to the cooperative behavioral tendencies described above, some personality traits may also exhibit social assortativity. Two of the "Big Five" personality traits—extraversion[11,12] and openness to experience[12]—appear to be more similar among friends than among individuals who are not friends with one another. However, the remaining Big Five traits do not predict friendship formation well[13]. Similarities in conscientiousness and neuroticism are not associated with friendship formation[12], and evidence for more similar levels of trait agreeableness among friends has been found in some studies[12], but not in others[11].

Thus, the extant research on homophily has recently begun to examine personality but has focused predominantly on demographic variables. It is possible that people cluster along these dimensions because they reflect commonalities in perceiving, thinking about, and reacting to the world. Similarity in how individuals interpret and respond to their environment increases the predictability of one another's thoughts and actions during social interactions[14], since knowledge about oneself is a more valid source of information about similar others than about dissimilar others. This increased predictability during social interactions, in turn, allows for less effortful and more confident communication, thus fostering more enjoyable social interactions, and increasing the likelihood of developing friendships[14]. In the same vein, interacting with individuals who share similar values, opinions, and interests may be rewarding because it reinforces one's own values, opinions, and interests, thus producing an implicit positive affective response, promoting attraction to similar others, and increasing the likelihood of developing friendships with individuals who see the world similarly to ourselves[15]. If friends are indeed exceptionally similar to one another in terms of how they perceive, interpret, and react to their environment, then social network proximity should be associated with similarity of cognitive processes as they unfold in real time. Whether or not humans tend to associate with others who see the world similarly has yet to be tested directly.

Here we tested the proposition that neural responses to naturalistic audiovisual stimuli are more similar among friends than among individuals who are farther removed from one another in a real-world social network. Measuring neural activity while people view naturalistic stimuli, such as movie clips, offers an unobtrusive window into individuals' unconstrained thought processes as they unfold[16]. Inter-subject correlations of neural response time series during natural viewing of complex, dynamic stimuli are associated with similarities in subjects' interpretation and understanding of those stimuli[16–19]. Thus, inter-subject similarities of neural response time series data afford insight into the similarity of individuals' thought processes as they experience the world around them. The current results suggest that neural response similarity decreases with increasing distance between individuals in their shared social network, such that friends have exceptionally similar neural responses. Social network proximity appears to be significantly associated with neural response similarity in brain regions involved in attentional allocation, narrative interpretation, and affective responding, suggesting that friends may be exceptionally similar in how they attend to, interpret, and emotionally react to their surroundings.

## Results

**Social network characterization**. We first characterized the social network of an entire cohort of students in a graduate program. All students ($N = 279$) in the graduate program completed an online survey in which they indicated the individuals in the program with whom they were friends (see Methods for further details). Given that a mutually reported tie is a stronger indicator of the presence of a friendship than an unreciprocated tie, a graph consisting only of reciprocal (i.e., mutually reported) social ties was used to estimate social distances between individuals. The same pattern of results to that described in our main analyses was observed when social distance was computed based on the presence of any reported social ties (i.e., when including unreciprocated social ties; Supplementary Note 1). The social network of the cohort is depicted in Fig. 1. Supplementary Fig. 1 illustrates the distribution of social distances between all dyads in the functional magnetic resonance imaging (fMRI) sample, as well as between all dyads in the entire cohort, and the degree distributions characterizing the fMRI study sample and entire cohort.

Using only mutually reported social ties yielded a network diameter of 6 for the entire cohort; using the existence of any social tie, irrespective of whether it was mutually reported, yielded a network diameter of 3. The density of the network, which is

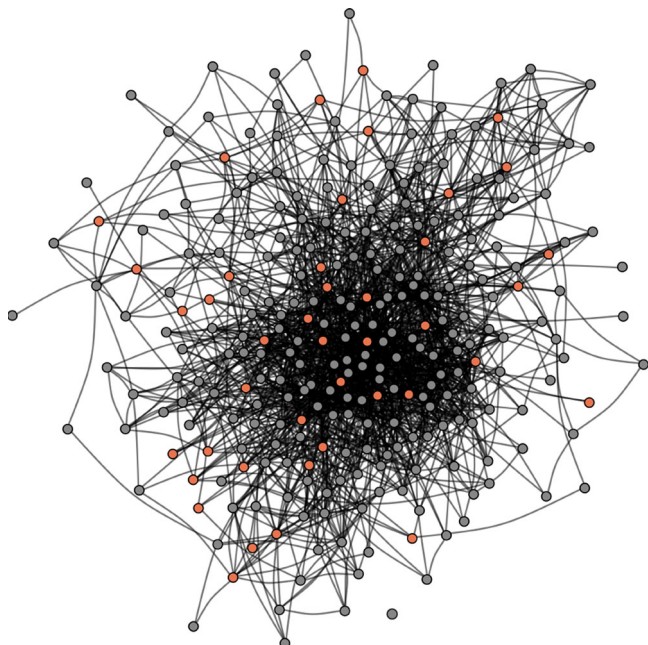

**Fig. 1** Social network. The social network of an entire cohort of first-year graduate students was reconstructed based on a survey completed by all students in the cohort ($N = 279$; 100% response rate). Nodes indicate students; lines indicate mutually reported social ties between them. A subset of students (orange circles; $N = 42$) participated in the fMRI study

defined as the ratio of the number of edges to the number of possible edges, excluding self-nominations, was 0.0451 when only including edges based on reciprocated social ties, and was 0.146 when establishing edges based on any social tie, including unreciprocated nominations. The total reciprocity of the graph, which refers to the probability that person$_i$ nominated person$_j$ as a friend if person$_j$ nominated person$_i$ as a friend, was 0.472, and the dyad-level rate of reciprocity, which refers to the probability of a mutually reported tie connecting members of a dyad, given the existence of any, possibly non-mutual, tie between them, was 0.309. Out-degree ranged from 2 to 146 ($M = 26.59$; SD = 23.33; median = 19), and in-degree ranged from 4 to 72 ($M = 26.59$; SD = 12.73; median = 24).

**Relating social network proximity to neural similarity**. A subset of students ($N = 42$) in the academic cohort described above participated in a fMRI study. During the fMRI study, each subject watched the same collection of video clips. The videos presented in the fMRI study covered a range of topics and genres (e.g., comedy clips, documentaries, and debates) that were selected so that they would likely be unfamiliar to subjects, effectively constrain subjects' thoughts and attention to the experiment (to minimize mind wandering), and evoke meaningful variability in responses across subjects (because different subjects attend to different aspects of them, have different emotional reactions to them, or interpret the content differently, for example). Prior to scanning, subjects were told that the video clips would vary in content and that their experience in the study would resemble watching television while someone else "channel surfed". All subjects experienced the same stimuli in the same order, and were provided with the same instructions. Therefore, differences in the similarities of subjects' neural response time courses likely stem from factors such as differences in subjects' dispositions, moods, cognitive styles, pre-existing assumptions, expectations, values, views, and interests, as well as differences in the pre-existing

knowledge structures into which incoming stimuli are integrated. We predicted that inter-subject similarities of neural responses among friends would be higher than among individuals who were farther removed from one another in the social network. Further, we tested whether similarities of neural responses can be used to predict the social distance between members of this social network.

Mean response time series spanning the course of the entire experiment were extracted from 80 anatomical regions of interest (ROIs) for each of the 42 fMRI study subjects (Methods; Fig. 2). For each of the 861 unique dyads in the sample, the Pearson correlation between the time series of fMRI responses was computed for each ROI. Pearson correlations were $z$-scored across dyads for each ROI prior to analysis and visualization in order to characterize the relative degree of synchrony in each dyad relative to other dyads for each brain region (Figs. 3 and 4). To test for a relationship between fMRI response similarity and social distance, a dyad-level regression model was used. Models were specified either as ordered logistic regressions with categorical social distance as the dependent variable or as logistic regression with a binary indicator of reciprocated friendship as the dependent variable. We account for the dependence structure of the dyadic data (i.e., the fact that each fMRI subject is involved in multiple dyads), which would otherwise underestimate the standard errors and increase the risk of type 1 error[20], by clustering simultaneously on both members of each dyad[21,22]. Cluster-robust standard errors account for both autocorrelation and possible heteroscedasticity in the data[21]; this method of accounting for dyadic dependence is comparable with approaches such as the quadratic assignment procedure or permutation testing[11].

For the purpose of testing the general hypothesis that social network proximity is associated with more similar neural responses to naturalistic stimuli, our main predictor variable of interest, neural response similarity within each student dyad, was summarized as a single variable. Specifically, for each dyad, a weighted average of normalized neural response similarities was computed, with the contribution of each brain region weighted by its average volume in our sample of fMRI subjects. (The same pattern of results was obtained when weighting each ROI equally, rather than in proportion to volume as described in Supplementary Note 3, or when neural response similarities were not normalized across subjects for each brain region prior to analysis, as described in Supplementary Note 2.) To account for demographic differences that might impact social network structure, our model also included binary predictor variables indicating whether subjects in each dyad were of the same or different nationalities, ethnicities, and genders, as well as a variable indicating the age difference between members of each dyad. In addition, a binary variable was included indicating whether subjects were the same or different in terms of handedness, given that this may be related to differences in brain functional organization[23]. All predictor variables were standardized to have a mean of 0 and a SD of 1 prior to analysis.

This model revealed a significant effect of neural similarity (ordered logistic regression: $\beta = -0.224$; SE = 0.105; $p = 0.03$; $N = 861$ dyads) on social distance that is striking in magnitude: holding other covariates constant, compared to a dyad at the mean level of neural similarity and at any given level of social distance, a dyad one SD more similar is 20% more likely to have social distance that is one unit shorter. Of the control variables also included in the model, differences between dyad members in terms of gender (ordered logistic regression: $\beta = 0.383$; SE = 0.122; $p = 0.002$; $N = 861$ dyads) and nationality (ordered logistic regression: $\beta = 0.561$; SE = 0.150; $p = 0.0002$; $N = 861$ dyads) were significantly related to social distance, whereas age (ordered

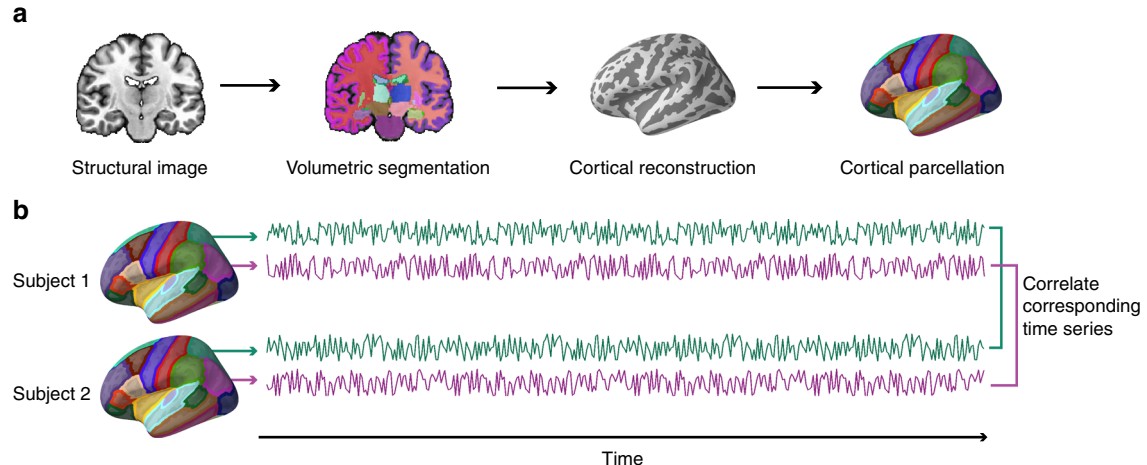

**Fig. 2** Computing inter-subject time series correlations. **a** Eighty anatomical regions of interest (ROIs) were derived for each individual using the FreeSurfer image analysis suite[53]. Segmentation of cerebral cortex, subcortical white matter, and deep gray matter volumetric structures (e.g., hippocampus, amygdala, and putamen) was performed on the high-resolution scan of each individual's brain volume. These structures are signified by color in the image demonstrating volumetric segmentation (e.g., the left and right cerebral cortex are shown in magenta and purple, respectively). Next, a cortical surface model was reconstructed and parcellated into anatomical units, which are signified by different colors in the cortical parcellation scheme illustrated on the far right. **b** For each individual, the average response time series within each ROI was extracted during video viewing. Next, the correlation between the time series extracted from each pair of corresponding ROIs was computed for each unique pair of subjects

logistic regression: $ß = 0.128$; SE = 0.137; $p = 0.35$), ethnicity (ordered logistic regression: $ß = 0.094$; SE = 0.095; $p = 0.32$; $N = 861$ dyads), and handedness (ordered logistic regression: $ß = 0.086$; SE = 0.060; $p = 0.15$; $N = 861$ dyads) were not (Fig. 5). To ascertain if neural similarity provided additional predictive power, above and beyond similarity in terms of the observed demographic variables, the full model described above was compared with a model that did not include neural similarity using a likelihood ratio test. Neural similarity added significant predictive power, above and beyond observable demographic similarity, $\chi^2(1) = 11.112$, $p = 0.0009$. A similar pattern of results was obtained when social distance was defined based on both reciprocated and unreciprocated social ties (Supplementary Note 1).

Logistic regressions that combined all non-friends into a single category, regardless of social distance, yielded similar results, such that neural similarity was associated with a dramatically increased likelihood of friendship, even after accounting for similarities in observed demographic variables. More specifically, a one SD increase in overall neural similarity was associated with a 47% increase in the likelihood of friendship (logistic regression: $ß = 0.388$; SE = 0.109; $p = 0.0004$; $N = 861$ dyads). Again, neural similarity improved the model's predictive power above and beyond observed demographic similarities, $\chi^2(1) = 7.36$, $p = 0.006$.

Results of analyses conducted separately for each video clip shown in the experiment are provided in Supplementary Table 3. We note that videos were presented in the same order to all subjects (to minimize inter-subject variability stemming from the manner in which clips were presented, rather than from endogenous differences between subjects) and that videos varied in duration (Table 1). Therefore, comparing results across video clips should be done with caution; these results are provided in case they are informative for future research.

To gain insight into what brain regions may be driving the relationship between social distance and overall neural similarity, we performed ordered logistic regression analyses analogous to those described above independently for each of the 80 ROIs, again using cluster-robust standard errors to account for dyadic dependencies in the data. This approach is analogous to common fMRI analysis approaches in which regressions are carried out independently at each voxel in the brain, followed by correction for multiple comparisons across voxels. We employed false discovery rate (FDR) correction to correct for multiple comparisons across brain regions. This analysis indicated that neural similarity was associated with social network proximity in regions of the ventral and dorsal striatum, including the right nucleus accumbens (ordered logistic regression: $ß = -0.231$; SE = 0.058; $p = 0.006$, FDR-corrected; $N = 861$ dyads), right caudate nucleus (ordered logistic regression: $ß = -0.279$; SE = 0.081; $p = 0.01$, FDR-corrected; $N = 861$ dyads), left caudate nucleus (ordered logistic regression: $ß = -0.231$; SE = 0.071; $p = 0.01$, FDR-corrected; $N = 861$ dyads), and left putamen (ordered logistic regression: $ß = -0.244$; SE = 0.071; $p = 0.01$, FDR-corrected; $N = 861$ dyads), the right amygdala (ordered logistic regression: $ß = -0.209$; SE = 0.064; $p = 0.01$, FDR-corrected; $N = 861$ dyads), the right superior parietal lobule (ordered logistic regression: $ß = -0.418$; SE = 0.121; $p = 0.01$, FDR-corrected; $N = 861$ dyads), and left inferior parietal cortex (ordered logistic regression: $ß = -0.385$; SE = 0.100; $p = 0.006$, FDR-corrected; $N = 861$ dyads). Regression coefficients for each ROI are shown in Fig. 6, and further details for ROIs that met the significance threshold of $p < 0.05$, FDR-corrected (two-tailed) are provided in Table 2.

To compare overall (i.e., weighted average) neural similarities across levels of social distance, Kolmogorov–Smirnov tests were used. Results indicated that average overall (weighted average) neural similarities were significantly higher among distance 1 dyads than dyads belonging to other social distance categories ($D = 0.19$, $p = 0.02$; $N = 861$ dyads). Distance 2 dyads were marginally more similar to one another than dyads in the other social distance categories ($D = 0.094$, $p = 0.06$; $N = 861$ dyads). Distance 3 dyads were significantly less similar than dyads in the other social distance categories ($D = 0.12$, $p = 0.004$; $N = 861$ dyads), and distance 4 dyads were not significantly different in overall neural response similarity from dyads in the other social distance categories ($D = 0.075$, $p = 0.67$; $N = 861$ dyads). All reported $p$-values are two-tailed.

To ensure that differences documented with Kolmogorov–Smirnov tests were due to differences in the location (rather than shape) of distributions, we conducted Wilcoxon rank-sum tests, which are specifically sensitive to the

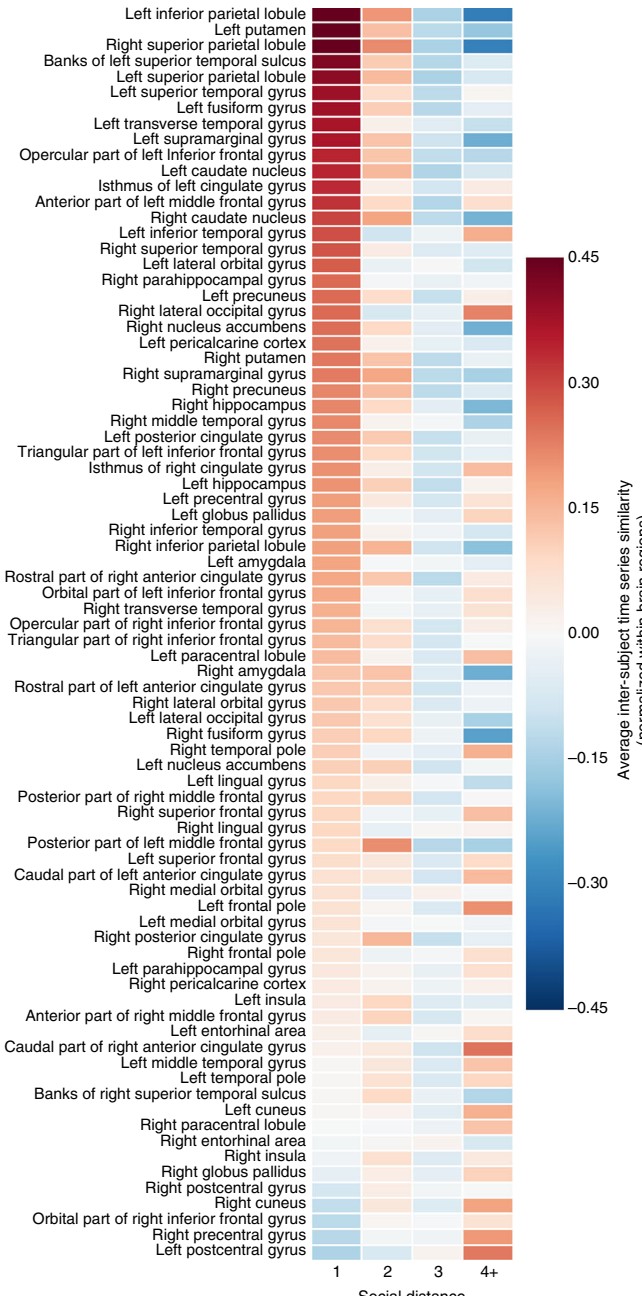

**Fig. 3** Inter-subject similarities for each brain region at each level of social distance. Inter-subject correlations of neural response time series for each of the 861 dyads were obtained for each of 80 anatomical regions of interest (ROIs). In order to illustrate how relative similarities of responses in each brain region varied as a function of social distance, inter-subject time series similarities (i.e., Pearson correlation coefficients between preprocessed fMRI response time series) were normalized (i.e., z-scored across dyads for each region) prior to averaging across dyads for each brain region within each social distance category. Warmer colors indicate relatively similar responses for a given brain region; cooler colors indicate relatively dissimilar responses for that brain region. Please note that because similarities have been normalized across dyads for each brain region, values depicted in this figure should be compared across social distance levels for each brain region, rather than across brain regions within or across social distances

difference in locations of two distributions, and which provided convergent results. Weighted average neural similarities were significantly higher among distance 1 dyads than among dyads from the other social distance categories ($W = 30\,570$, $p = 0.004$; $N = 861$ dyads). The same was true for distance 2 dyads ($W = 91\,356$, $p = 0.008$; $N = 861$ dyads). Distance 3 dyads were less similar overall than dyads belonging to the other social distance categories ($W = 79\,062$, $p = 0.0002$; $N = 861$ dyads). Distance 4 dyads did not differ significantly from dyads in the remaining social distance categories ($W = 36\,918$, $p = 0.63$; $N = 861$ dyads). Pairwise analyses suggested that distance 1 dyads were significantly more similar to one another than distance 3 ($W = 16\,598$, $p = 0.00036$; $N = 475$ dyads) and distance 4 ($W = 3856$, $p = 0.016$; $N = 163$ dyads) dyads, but not distance 2 dyads ($W = 10\,116$, $p = 0.13$; $N = 349$ dyads). Distance 2 dyads were more similar than distance 3 dyads ($W = 67\,759$, $p = 0.00074$; $N = 698$ dyads). Perhaps reflecting the large variability among distance 4 dyads, distance 4 dyads did not differ significantly from distance 2 ($W = 15\,695$, $p = 0.15$; $N = 386$ dyads) or distance 3 dyads ($W = 19\,631$, $p = 0.47$; $N = 512$ dyads). All reported $p$-values are two-tailed. Permutation tests that involved randomly shuffling fMRI data across subjects while holding the topological structure of the network connecting subjects constant provided convergent results, as described in Supplementary Note 5 and Fig. 7.

Figures 3 and 4a–c illustrate the average relative level of neural similarity among dyads within each social distance category for each individual brain region. In order to illustrate how overall neural similarity varies as a function of social distance while holding all control variables (i.e., handedness, age, gender, ethnicity, and nationality) constant, deviation-coded point estimates were computed and are illustrated in Fig. 4d. Deviation coding provides, for each social distance, a point estimate and confidence interval of the difference in neural similarity from the average of the other social distance categories; complete details appear in the Supplementary Methods.

**Out-of-sample prediction.** We also tested whether it was possible to predict friendship status based on similarity of fMRI response time series across brain regions. If so, it should be possible to build a predictive model of social distance by training an algorithm to recognize patterns of neural similarities associated with various social distance categories from a subset of dyads' data. This model should then correctly generalize to predicting the social distances characterizing new dyads.

Eighty-element vectors of neural similarities were extracted for all 861 dyads of fMRI subjects. Given that the current data set is imbalanced across social distance categories (e.g., there are far fewer distance 1 dyads than distance 3 dyads), data resampling and folding procedures were used to create a series of balanced training and testing data sets such that all dyads were included in analyses (see Methods for further details). Within the training data set for each data fold, a grid search procedure[24] was used to select the C parameter of a linear support vector machine (SVM) learning algorithm that would best separate dyads according to social distance. Following hyper-parameter tuning, the classifier was trained on the entire training data set within a given data fold to predict the social distances characterizing dyads based on corresponding patterns of inter-subject neural time course similarity. Finally, the predictive performance of this classifier was evaluated on data from the testing data set within the data fold, which was comprised of data from dyads to which the model had not previously been exposed. This procedure was performed for each data fold, and then cross-validated predictive performance was averaged across data folds (see Methods for further details).

As shown in Fig. 8, the classifier tended to predict the correct social distances for dyads in all distance categories at rates above the accuracy level that would be expected based on chance alone (i.e., 25% correct), with an overall classification accuracy of 41.25%. Classification accuracies for distance 1, 2, 3, and 4 dyads were 48%, 39%, 31%, and 47% correct, respectively. As illustrated in the confusion matrix in Fig. 8a, for all social distance categories, the correct distance label was predicted most often, with confusions (i.e., incorrect predictions) occurring most frequently in columns adjacent to the elements along the diagonal. The latter pattern of results reflects the fact that in

cases where the classifier assigned the incorrect social distance label to a dyad, it tended to be only one level of social distance away from the correct answer: when friends were misclassified, they were misclassified most often as distance 2 dyads; when distance 2 dyads were misclassified, they were misclassified most often as distance 1 or 3 dyads, and so on. Permutation testing was performed in order to test if overall cross-validated classification accuracy significantly exceeded chance. Specifically, the distribution of classification accuracies that would be achieved based on chance alone was obtained by repeating the classification analysis after having randomly shuffled the distance category labels in the

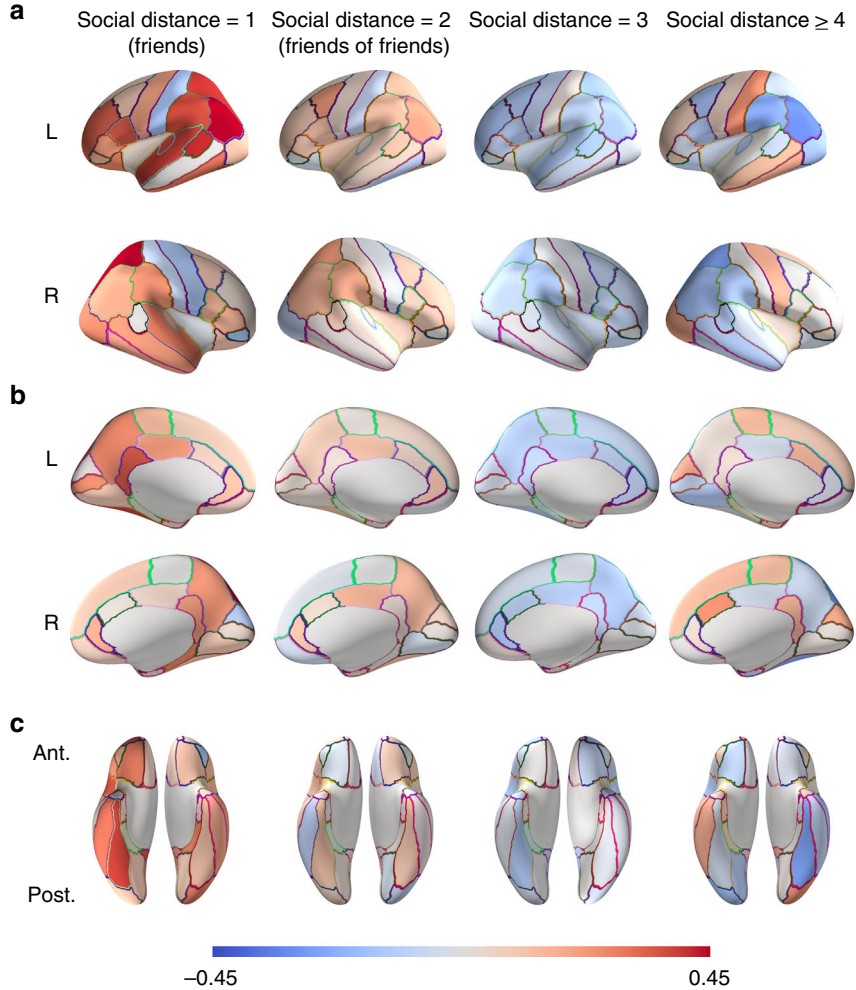

training data 1000 times. The results of this permutation testing procedure are visualized in Fig. 8b, and suggest that the overall classification accuracy was significantly higher than what would be expected based on chance, $p = 0.004$ ($N = 861$ dyads; see Methods for further details).

## Discussion

The results reported here are consistent with neural homophily: people tend to be friends with individuals who see the world in a similar way. Neural responses during unconstrained viewing of movie clips were significantly more similar among friends than among people farther removed from one another in their real-world social network. More generally, people who responded more similarly to the videos shown in the experiment were more likely to be closer to one another in their shared social network, and these effects were significant even when controlling for inter-subject similarities in demographic variables, such as age, gender, nationality, and ethnicity. In addition, predictive models trained to discern social distance based solely on patterns of inter-subject neural response similarity were able to accurately generalize to novel data, correctly predicting the friendship status and social distance of new pairs of individuals based only on those dyads' patterns of neural response similarities.

Much previous research has shown that humans tend to associate with others who are similar to themselves in terms of a wide range of characteristics, including demographic information (e.g., age, gender, and ethnicity)[2], certain personality traits and behavioral tendencies[11,12], and even aspects of our genotypes[25,26]. The current findings extend this research by demonstrating that covert mental responses to the environment, as indexed by neural processes evoked naturalistically during undirected viewing of videos, are exceptionally similar among friends.

Brain areas where response similarity was associated with social network proximity included subcortical areas implicated in motivation, learning, affective processing, and integrating information into memory, such as the nucleus accumbens, amygdala, putamen, and caudate nucleus[27–29]. Social network proximity was also associated with neural response similarity within areas involved in attentional allocation, such as the right superior parietal cortex[30,31], and regions in the inferior parietal lobe, such as the bilateral supramarginal gyri and left inferior parietal cortex (which includes the angular gyrus in the parcellation scheme used[32]), that have been implicated in bottom-up attentional control, discerning others' mental states, processing language and the narrative content of stories, and sense-making more generally[33–35]. Many of these regions have previously been demonstrated to become tightly coupled when subjects are similarly emotionally engaged, such as the amygdala, ventral striatum, and inferior parietal cortex[36]; when people are provided with shared

contexts for understanding a situation, such as the inferior parietal lobe in the vicinity of the temporoparietal junction[33]; or when people adopt similar psychological perspectives, such as the superior and inferior posterior parietal cortex[37]. We hesitate to make strong inferences about the specific mental processes that underlie the results observed here, given that many of these regions are functionally heterogeneous. However, the current results suggest that social network proximity may be associated with similarities in how individuals attend to, interpret, and emotionally react to the world around them.

We did not directly compare the results obtained in the current study to those that might be obtained by using behavioral measures, such as explicit questions about subjects' reactions to experimental stimuli, or self-report measures of individual difference variables. Therefore, we cannot ascertain if comparable results could have been achieved without the use of neuroimaging. That said, we suggest that the paradigm used here offers several benefits compared with other methods of assessing similarities in how individuals respond to their environment. First, the rich, engaging, and dynamic stimuli used likely recruit a relatively large proportion of the emotional and cognitive processes that characterize everyday mental life, and do so unobtrusively in a relatively ecologically valid manner[38]. This is beneficial not only because it allows subjects' mental processes to unfold without interruption; it also allows for the neural processes underlying such processes to be measured contemporaneously, as they transpire, rather than asking subjects to reflect on those processes after they occur and report on those reflections to experimenters. A large body of social psychological literature has demonstrated that our ability to accurately introspect about our own mental processes is often limited[39]. We appear to lack conscious access to many aspects of mental processing[40], limiting the efficacy of self-report measures for capturing many psychological phenomena. In contrast, neuroimaging facilitates the measurement of aspects of mental processing to which we lack conscious access, but that nevertheless impact behavior[41]. Similarly, compared with self-report, the validity of responses obtained using the current paradigm is less likely to be threatened by subjects' attempts to present themselves in a socially desirable manner, which can distort experimental results in a variety of ways[42]. In addition, measuring fMRI responses from the whole brain simultaneously confers the benefit of concurrently measuring brain activity associated with diverse aspects of mental processing. Rather than being limited to a few targeted questions, using data recorded from the entire brain during natural viewing allows for neural processing to be captured associated with whatever emotional (e.g., amusement, disgust, sadness, desire, and fear) and cognitive (e.g., attention to different aspects of the stimulus; interpretations of a video as they are informed by subjects' pre-existing assumptions, knowledge, and values; and

**Fig. 4** Inter-subject similarities by social distance. **a–c** Average dyadic fMRI response time series similarities overlaid on a cortical surface model (**a** lateral view; **b** medial view; **c** ventral view). In order to illustrate how relative similarities of responses in each brain region varied as a function of social distance, inter-subject time series similarities (i.e., Pearson correlation coefficients between preprocessed fMRI response time series) were normalized (i.e., z-scored across dyads for each region) prior to averaging across dyads for each brain region and overlaying results on an inflated model of the cortical surface for each social distance category. Warmer colors indicate relatively similar responses for a given brain region; cooler colors indicate relatively dissimilar responses for that brain region. Please note that because similarities have been normalized across dyads for each brain region, values depicted in this figure should be compared across social distance levels for each brain region, rather than across brain regions within or across social distances. Please see Fig. 3 for presentation of results that include subcortical gray matter structures. Ant. = anterior; Post. = posterior; L = left; R = right. **d** Deviation-coded point estimates and 95% CIs for weighted average neural similarities, after accounting for inter-subject similarities in control variables (demographic variables and handedness) are shown for distance 1 (deviation-coded point estimate = 0.23, 95% CI [0.07, 0.41]), distance 2 (deviation-coded point estimate = 0.03, 95% CI [−0.11, 0.17]), distance 3 (deviation-coded point estimate = −0.20, 95% CI [−0.30, −0.09]), and distance 4 (deviation-coded point estimate = −0.07, 95% CI [−0.29, 0.14]) dyads. Deviation coding measures the difference in overall neural similarity between dyads within each social distance category and the average overall neural similarity of dyads in the other social distance categories, after removing the effects of control variables. For further details on deviation coding, please refer to the Supplementary Methods. Cortical surface visualizations were created using PySurfer[58]

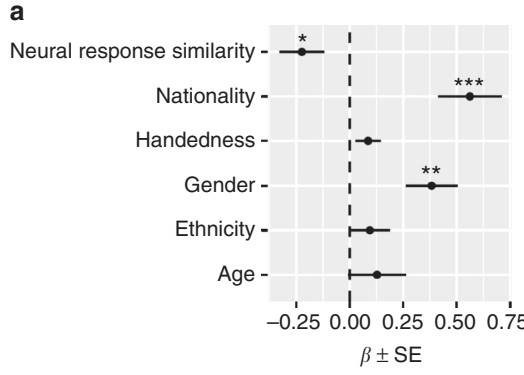

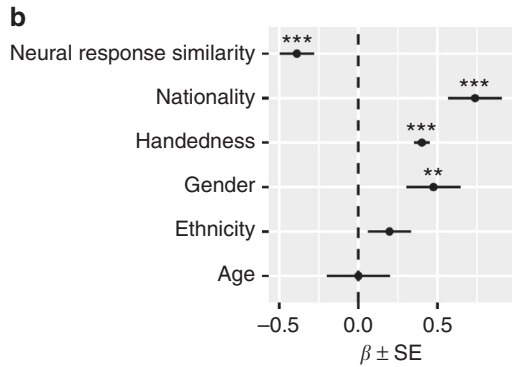

**Fig. 5** Regression coefficients from models predicting social distance and friendship status. Regression coefficients correspond to weighted average neural response similarities and dissimilarities in control variables. **a** Illustration of regression coefficients from an ordered logistic regression model in which social distance was predicted based on the similarity of participants' neural response time series, as well as dissimilarities in control variables. **b** Illustration of regression coefficients from a logistic regression model in which friendship status was predicted based on the similarity of participants' neural response time series, as well as dissimilarities in control variables. Error bars indicate standard errors of the regression coefficients estimated using multi-way clustering to account for dyadic dependencies in the data set. ***$p < 0.001$, **$p < 0.01$, *$p < 0.05$

waxing and waning levels of overall attentional engagement) responses happen to be elicited, at whatever time points those responses happen to be recruited. Even if it was possible to assess the same information using self-report questionnaires, it would presumably be necessary to use an extremely large battery of questions in order to do so.

On the other hand, while the naturalistic neuroimaging paradigm used here confers many advantages, a more specific understanding of precisely which cognitive and emotional processes underlie these effects will likely require complementary follow-up studies involving behavioral measures and more constrained experimental paradigms. In addition, a single sequence of stimuli was used for the current study in order to provide a common context throughout all time points in the experiment for all subjects. Future studies may wish to adopt experimental designs that allow for drawing inferences about exactly what kinds of stimuli are particularly important for predicting patterns of real-world social ties.

Interestingly, although increased distance between individuals in the social network was associated with decreased neural response similarity overall, the level of neural response similarity among distance 4 dyads was highly variable and did not differ significantly from that of distance 2 or 3 dyads. There are at least two reasons why the pattern of results observed up to a distance of three may have dissipated at distance 4. First, it is possible that

individuals at distances greater than three simply do not encounter one another frequently enough to have the opportunity to become friends. Therefore, the collection of dyads characterized by a social distance of four or more may include some dyads that would be compatible and others that would be incompatible as friends. A second, not mutually exclusive, possibility pertains to the "three degrees of influence rule" that governs the spread of a wide range of phenomena in human social networks[43]. Data from large-scale observational studies as well as lab-based experiments suggest that wide-ranging phenomena (e.g., obesity, cooperation, smoking, and depression) spread only up to three degrees of geodesic distance in social networks, perhaps due to social influence effects decaying with social distance to the extent that the they are undetectable at social distances exceeding three, or to the relative instability of long chains of social ties[43]. Although we make no claims regarding the causal mechanisms behind our findings, our results show a similar pattern.

Do we become friends with people who respond to the environment similarly, or do we come to respond to the world similarly to our friends? Although the results of the current study suggest that friends have exceptionally similar neural responses to naturalistic stimuli, due to this study's cross-sectional nature, we cannot ascertain, based on these results alone, whether neural response similarity is a cause or consequence of friendship. Thus, future longitudinal studies should measure whether inter-subject neural response similarities predict subsequent friendship formation among members of evolving social networks. We anticipate that such studies will find that the exceptional similarity of neural responses among friends reflects both homophily and social influence processes. A large body of research demonstrates that people in our immediate environment influence how we think, feel, and behave[44,45], and humans' embeddedness within social networks causes these social influence effects to reverberate outward in social ties, and thus, to extend beyond those individuals with whom we interact with directly[46]. At the same time, similar people may tend to become connected at higher rates because they find themselves in common situations[47]. Similarly, pre-existing similarities in how individuals tend to perceive, interpret, and respond to their environment can enhance social interactions and increase the probability of developing a friendship via positive affective processes and by increasing the ease and clarity of communication[14,15]. Future research should extend the current findings by adopting longitudinal experimental designs that afford insight into the extent to which the results observed here reflect homophily, social influence processes, or a combination of these phenomena.

In summary, the current results suggest that friends are exceptionally similar to one another in terms of how they perceive, interpret, and react to the world around them, as reflected in unobtrusive measurements of mental processes as they unfold over time. Proximity in terms of social ties in a real-world social network was associated with similarity in fMRI response time series in brain regions implicated in attending to and interpreting the sensory environment, as well as emotional responding. These data also demonstrate that it is possible to predict whether or not two individuals are friends, as well as more nuanced social distance information (i.e., geodesic distance in a real-life social network) based only on the similarity of temporal patterns in their neural responses during free viewing of complex, real-world scenes. Time courses of individuals' neural responses to continuous, naturalistic stimuli provide information-rich signatures of those individuals' responses to the stimuli, which are presumably shaped by characteristics of those individuals' dispositions, pre-existing knowledge, views, opinions, interests, and values. These signatures can be used to identify individuals who are likely to be friends, as well as individuals who are likely to be

**Table 1 Summary of video clips shown in the fMRI study**

| Clip | | Description | Duration (s) |
|---|---|---|---|
| 1 | 'An Astronaut's View of Earth' | An astronaut discusses viewing Earth from space, and in particular, witnessing the effects of climate change from space. He then urges viewers to mobilize to address this issue | 223 |
| 2 | Google Glass review | A journalist wears a Google Glass headset for a day and weighs the pros and cons of being an 'early adopter' of this technology | 88 |
| 3 | 'Crossfire' | Two journalists debate the appropriateness of President Obama's use of humor in a speech; excerpts from the speech are shown | 89 |
| 4 | 'All I Want' | A sentimental music video depicting a social outcast with a facial deformity seeking companionship | 305 |
| 5 | Wedding film | A homemade film depicting scenes from two men's wedding ceremony and subsequent celebration with family and friends | 120 |
| 6 | Scientific demonstration | An astronaut at the International Space Station demonstrates and explains what happens when one wrings out a waterlogged washcloth in space | 118 |
| 7 | 'Food Inc.' | An excerpt from a documentary discussing how the fast food industry influences food production and farming practices in the United States | 178 |
| 8 | 'We Can Be Heroes' | An excerpt from a mockumentary-style series in which a man discusses why he nominated himself for the title of Australian of the Year | 202 |
| 9 | 'Ban College Football' | Journalists and athletes debate whether or not football should be banned as a college sport | 195 |
| 10 | Soccer match | Highlights from a soccer match | 91 |
| 11 | Baby sloth sanctuary | A documentary about caring for baby sloths at a sanctuary in Costa Rica | 200 |
| 12 | 'Ew!' | A comedy skit in which grown men play teenage girls disgusted by things around them | 169 |
| 13 | 'Life's Too Short' | An example of 'cringe comedy' in which a dramatic actor is depicted unsuccessfully trying his hand at improvisational comedy | 106 |
| 14 | 'America's Funniest Home Videos' | A series of homemade video clips depicting examples of unintentional physical comedy arising from accidents | 101 |

indirectly connected via mutual friends, in a real-world social network.

## Methods

**Social network characterization.** Subjects in part 1 of the study (social network characterization) were 279 (89 females) first-year students in a graduate program at a private university in the United States who participated as part of their course-work on leadership. The total size of the graduate cohort was 279 students (i.e., all students in the cohort participated in the leadership course); a 100% response rate was obtained for part 1 of the study, which was done in accordance with the standards of the local ethical review board. The social network survey was administered during November of students' first academic year in the graduate program, which began the preceding August. Therefore, subjects had been on campus together for 3–4 months prior to completing the social network survey, and friendships reported on the survey would have been formed either during subjects' first few months on campus or prior to entering the graduate program.

In order to characterize the social network of all first-year students, an online social network survey was administered. Subjects followed an e-mailed link to the study website where they responded to a survey designed to assess their position in the social network of students in their cohort of the academic program. The survey question was adapted from Burt[48] and has been previously used in the modified form used here[11,49,50]. It read, "Consider the people with whom you like to spend your free time. Since you arrived at [institution name], who are the classmates you have been with most often for informal social activities, such as going out to lunch, dinner, drinks, films, visiting one another's homes, and so on?" A roster-based name generator was used to avoid inadequate or biased recall. Classmates' names were listed in four columns, with one column corresponding to each section of students in the graduate program. Students' names were listed alphabetically within section. Subjects indicated the presence of a social tie with an individual by placing a checkmark next to his or her name. Subjects could indicate any number of social ties, and had no time limit for responding to this question. The social network of the cohort is illustrated in Fig. 1. The social network survey used here only inquired about students' interactions with other members of their academic cohort. Subjects undoubtedly have interactions with individuals outside of their cohort of classmates that this survey did not measure (e.g., with family members, prior colleagues, friends from before they entered the program, etc.). We note that the current study was conducted at a relatively small and remotely located institution where subjects' contacts outside of campus likely play a smaller role in their daily lives compared to their quotidian, face-to-face interactions with their classmates. That said, social distances between some subjects who did not report friendships with one another may be underestimated due to indirect connections through individuals outside of the graduate cohort.

In addition, demographic data about each subject's gender, ethnic identity, and country of citizenship were obtained from the school's registrar. Personally identifying information was removed from these data; subjects' demographic, social network, and neuroimaging data were linked only by anonymous ID numbers.

Social network analysis was performed using the R package igraph[51,52]. An unweighted, undirected graph consisting only of reciprocal (i.e., mutually reported) social ties was used to estimate social distances between individuals. For example, an undirected edge would connect two actors, person$_i$ and person$_j$, only if person$_i$ and person$_j$ each nominated the other as a friend. If person$_i$ nominated person$_j$, but person$_j$ did not nominate person$_i$, or vice versa, these actors were not considered friends for the purposes of this study. Social distance was operationalized as the smallest number of intermediary, mutual social ties required to connect two individuals in the network (i.e., geodesic distance). Pairs of individuals who both named one another were assigned a social distance of one. An individual would be assigned a distance of two from a given subject if he or she had a mutually reported friendship with that subject's friend, but not with the subject him or herself, and so on. The distribution of social distances for all pairs of fMRI study subjects is provided in Supplementary Fig. 1.

**fMRI study subjects.** Forty-two subjects (12 female; 3 left-handed) aged 25–32 ($M = 27.98$; SD = 1.72) who had completed part 1 of the study completed a subsequent neuroimaging study (part 2). Students were informed during class about the opportunity to participate in an fMRI study involving viewing visual stimuli. They were informed that they would receive $20 per hour as compensation for their time, as well as anatomical images of their brains. All students who were interested in participating and were not affected by standard safety-related contraindications for MRI (e.g., the presence of metallic implants) participated in the neuroimaging study. All subjects were fluent in English and had normal or corrected-to-normal vision. Because subjects were not allocated to experimentally defined groups in the current study, blinding investigators to between-subject conditions and random assignment of subjects to conditions were not applicable. Subjects provided informed consent in accordance with the policies of the local ethical review board. Data collection for the neuroimaging study began mid-way through February during subjects' first academic year in the graduate program, and all scanning was completed within 2 weeks. Therefore, all neuroimaging data was collected ~3 months after the collection of the social network data.

**fMRI data acquisition.** Subjects were scanned using a 3 T Philips Achieva Intera scanner with a 32-channel head coil. An echo-planar sequence (35 ms echo time (TE); 2000 ms repetition time (TR); 3.0 mm × 3.0 mm × 3.0 mm resolution; 80 × 80 matrix size; 240 × 240 mm field of view (FOV); 35 interleaved transverse slices with no gap; 3.0 mm slice thickness) was used to acquire functional images. Stimuli were presented over the course of six functional runs. Functional runs consisted of 204, 276, 194, 147, 189, and 108 dynamic scans, for a total functional data acquisition time of approximately 33.7 min, excluding time between functional runs. A high-resolution T1-weighted anatomical scan was also acquired for each subject (8.2 s TR; 3.7 ms TE; 240 × 187 FOV; 0.938 mm × 0.938 mm × 1.0 mm resolution) at the end of the scanning session. Foam padding was placed around subjects' heads to minimize head motion.

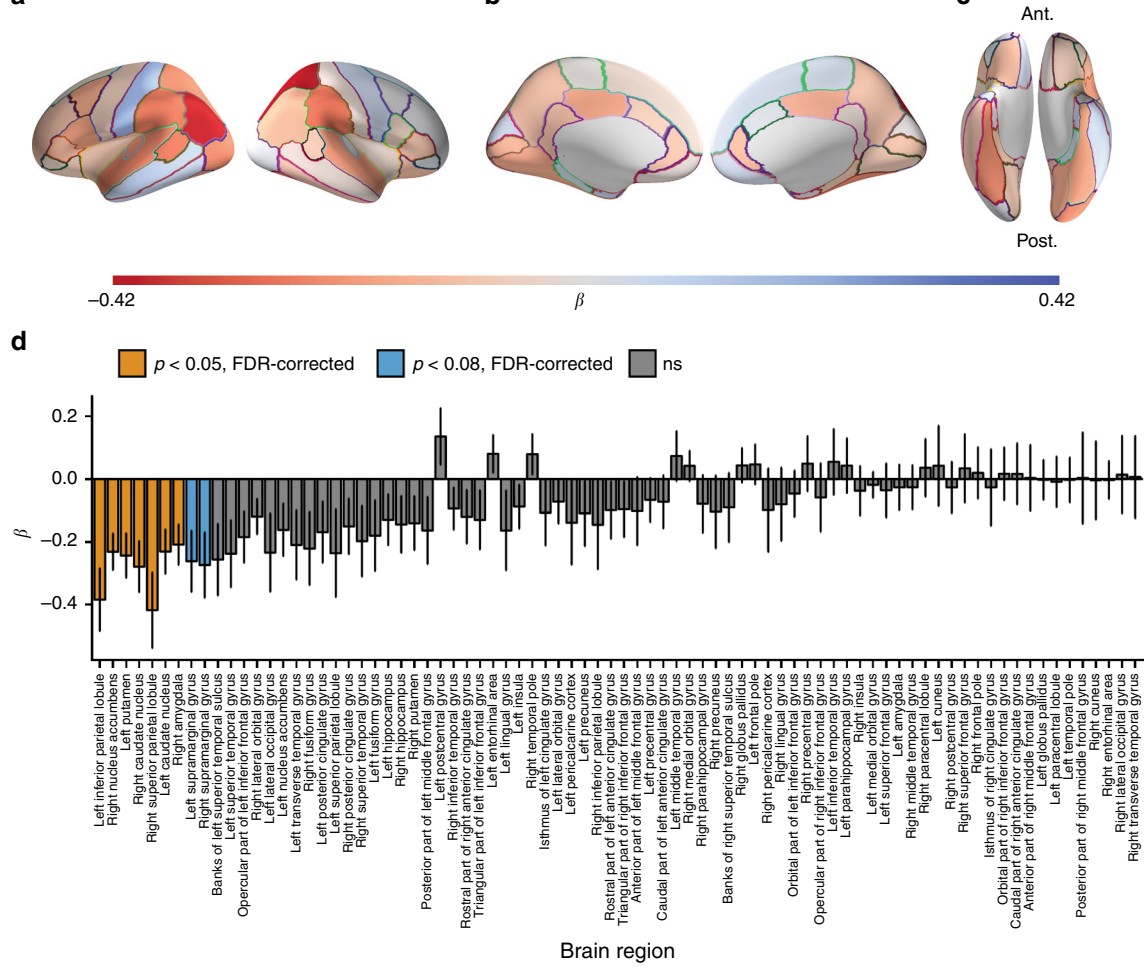

**Fig. 6** Testing associations between neural response similarity and social distance by brain region. As described in the main text, ordered logistic regression analyses were carried out for each brain region in which social network distances were modeled as a function of local neural response similarities and dyadic dissimilarities in control variables (gender, ethnicity, nationality, age, and handedness). Negative regression coefficients for neural response similarity indicate that greater neural response similarity was associated with decreased social distance. Regression coefficients for the effects of neural response similarity on social distance for each cortical ROI are shown overlaid on **a** lateral, **b** medial, and **c** ventral views of the cortical surface. Ant. = anterior; Post. = posterior. In each view, the left hemisphere is displayed on the left. Cortical surface visualizations were created using PySurfer[58]. Warmer colors indicate negative regression coefficients (i.e., where greater neural response similarity was associated with social network proximity), whereas cooler colors indicate positive regression coefficients (i.e., where greater neural response similarity was associated with increased social distance). **d** Regions where neural similarity was significantly predictive of social distance, above and beyond the effects of control variables ($p < 0.05$, FDR-corrected, two-tailed) are shown in yellow, with marginally significant regions ($p < 0.08$) shown in blue, and all other regions shown in gray. Error bars indicate cluster-robust standard errors of the regression coefficients

**Table 2 Brain regions where neural similarity was significantly predictive of social distance above and beyond similarities in control variables**

| Hemi | Region | β | SE | *p*-value (uncorrected) | FDR-corrected *p*-value |
|---|---|---|---|---|---|
| R | Nucleus accumbens | −0.23 | 0.058 | 0.000077 | 0.0055 |
| L | Inferior parietal cortex | −0.38 | 0.10 | 0.00014 | 0.0055 |
| R | Superior parietal cortex | −0.42 | 0.12 | 0.00056 | 0.011 |
| R | Caudate nucleus | −0.28 | 0.081 | 0.00064 | 0.011 |
| L | Putamen | −0.24 | 0.071 | 0.00068 | 0.011 |
| L | Caudate nucleus | −0.23 | 0.071 | 0.0011 | 0.014 |
| R | Amygdala | −0.21 | 0.064 | 0.0012 | 0.014 |
| *L* | *Supramarginal gyrus* | *−0.26* | *0.098* | *0.0076* | *0.076* |
| *R* | *Supramarginal gyrus* | *−0.27* | *0.10* | *0.0090* | *0.076* |

Regions where neural similarity was significantly predictive of social distance ($p < 0.05$, FDR-corrected, two-tailed) are shown; marginally significant results ($p < 0.08$) are italicized. Ordered logistic regression analyses were carried out for each brain region using multi-way clustering to account for dyadic dependencies in the data. Negative regression coefficients indicate that greater neural response similarity was associated with decreased social distance
FDR false discovery rate, Hemi hemisphere, L left, R right

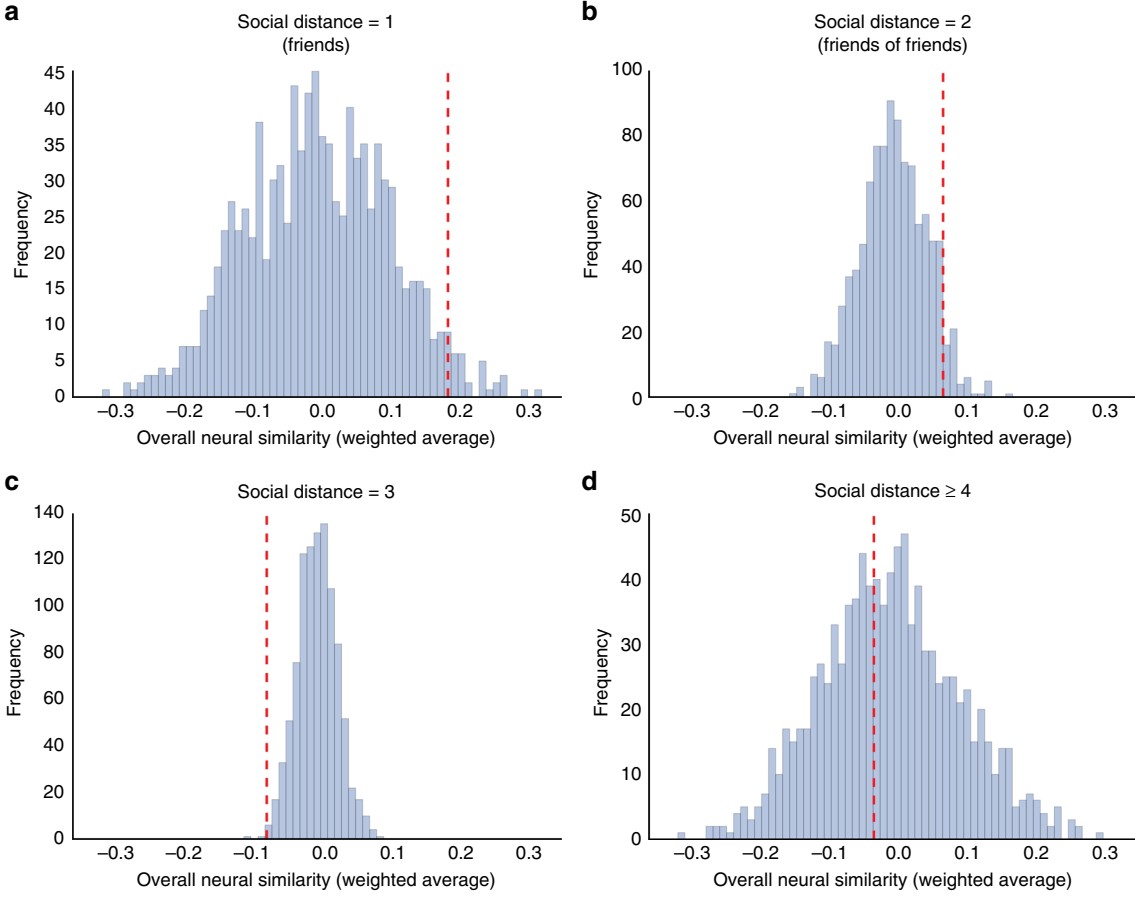

**Fig. 7** Results of permutation testing based on network randomization. Histograms depict the distribution of average overall neural similarities for **a** distance 1, **b** distance 2, **c** distance 3, and **d** distance 4 dyads across 1000 permutations of the data set in which fMRI responses were randomly shuffled across participants while the topological structure of the network of social connections between those participants was held constant. Red dashed lines depict the actual neural response similarities (i.e., based on the non-permuted data) for dyads corresponding to each social distance category. Results of these permutation tests indicated that distance 1 dyads ($N = 63$) were more similar than would be expected based on chance ($p = 0.03$), distance 2 dyads ($N = 286$) were marginally more similar than would be expected based on chance ($p = 0.06$), and distance 3 dyads ($N = 412$) were less similar to one another than would be expected based on chance ($p = 0.003$). Distance 4 dyads ($N = 100$) were neither more nor less similar to one another than would be expected if there were no relationship between overall neural response similarity and proximity in the social network ($p = 0.5$)

**fMRI study paradigm.** Prior to being scanned, subjects were informed that they would be watching a series of videos while in the scanner. Subjects were informed that these videos would be brief and would vary in content, and that the experience of participating in the study would be analogous to passively watching television while someone else "channel surfed." Videos were presented in the same order to all subjects in order to avoid inducing inter-subject response variability that would be attributable simply to differences in the manner in which clips were presented in the experiment (e.g., if a serious video happened to be preceded by a comedic clip for some subjects and not others). Given that the current study aimed to test if subjects' positions relative to one another in their social networks are associated with neural response similarity, rather than to contrast responses to particular stimuli, the benefits of using a single trial order for all subjects were judged to outweigh potential costs. After the scanning session had concluded, the experimenter interviewed each subject to determine if he or she had previously seen any of the video clips used in the experiment.

**fMRI study stimuli.** Stimuli consisted of 14 videos presented with sound over the course of six fMRI runs. Videos ranged in duration from 88 to 305 s (Table 1). Three principal criteria were used to select video clips as stimuli. First, we sought to select stimuli that subjects in our sample would be relatively unlikely to have seen before. This was done in order to avoid inducing differences in inter-subject correlations due to simple familiarity with the stimuli, given that friends may be more likely to have seen the same videos prior to the experiment compared with pairs of individuals who are not friends with one another.

Second, we sought to select engaging stimuli. We reasoned that insufficiently engaging stimuli would be likely to evoke mind wandering, which would likely involve idiosyncratic thoughts unrelated to the experiment, and thus would introduce unwanted noise into estimates of inter-subject correlations and their

relationships to social distance. In contrast, stimuli that effectively engage an audience do so by directing and constraining viewers' thoughts and associated neural activity. As such, professionally directed movies and television shows elicit more reliable responses within and across subjects than unedited video footage or series of static photographs[38]. Such videos are engineered to engage viewers' attention and drive their inferences by inducing particular reactions and interpretations at specific times, and thus, are well-suited for experiments seeking to induce a shared series of cognitive states across subjects[18].

Third, we sought to select stimuli that, while engaging, would also introduce meaningful variability in inter-subject correlations. We reasoned that for the purposes of the current study, uninformative inter-subject variability in neural response time series data would arise largely from using stimuli that failed to effectively engage subjects, and thus, failed to constrain their thoughts and attention. In contrast, meaningful inter-subject variability in neural response time series data would arise from using stimuli that produced diverging inferences and patterns of attentional allocation in different sets of viewers. We sought to select stimuli that minimized uninformative inter-subject variability by engaging subjects' attention, but at the same time, promoted meaningful inter-subject variability by evoking divergent reactions across subjects. For example, videos were chosen that might be interpreted as sweet by some subjects, but cloying or "sappy" by others (e.g., a sentimental music video), that would appeal to different styles of humor (e.g., physical comedy, wry humor, "cringe" comedy, and sophomoric or "lowbrow" humor), and that presented one or both sides of an argument that subjects might resonate with or respond to with criticism (e.g., a debate about whether college football should be banned). Brief descriptions of all 14 videos are presented in Table 1.

The majority of subjects (29 of 42) had not seen any of the video clips used in the fMRI study prior to participating, and the average number of clips subjects had seen before was low ($M = 0.41$ clips out of 14; SD = 0.70). For the majority of

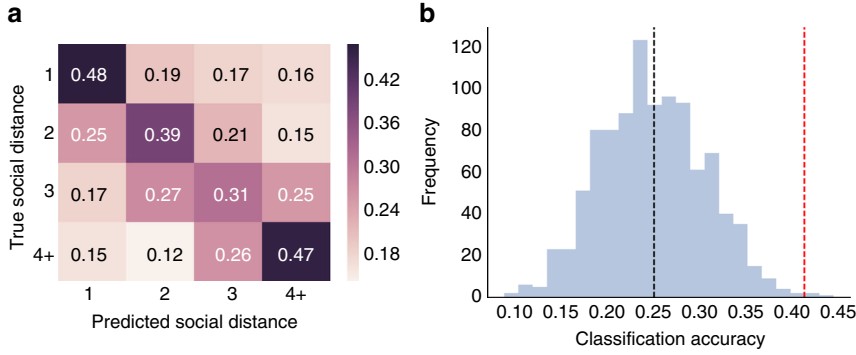

**Fig. 8** Predicting social distance based on inter-subject neural similarities. **a** Confusion matrix summarizing cross-validated prediction accuracy of four-way classifiers trained to predict the geodesic distance between members of dyads in their social network based on patterns of neural similarity, averaged across data folds (see Methods for further details). Numbers and cell colors indicate how often the classifier predicted that dyads belonged to each social distance category (chance = 0.25). **b** Permutation testing was used to compare the overall cross-validated prediction accuracy to random chance. The distribution of accuracies achieved by repeating the classification analyses after randomly shuffling the data category labels in the training folds 1000 times is shown in blue; the black dashed line depicts the average level of accuracy achieved in the randomly permuted data. The red dashed line indicates the actual overall cross-validated classification accuracy, which significantly exceeded chance ($p = 0.004$)

videos used as experimental stimuli (i.e., 9 of 14), there were no dyads whose members had both seen the clip prior to scanning. Of the remaining video clips, two had previously been seen by two subjects (i.e., by both members of a single dyad, or 0.12% of all dyads), two had been seen previously by three subjects (i.e., by both members of three dyads, or 0.35% of dyads), and one clip had been seen previously by four subjects (i.e., by both members of six dyads, or 0.70% of the 861 total dyads). Please refer to Supplementary Table 1 for a complete summary of subjects' reported familiarity with the 14 video clips used as experimental stimuli, and Supplementary Note 4 for a replication of our main analyses excluding dyads whose members had both seen any of the same stimuli prior to participating in the fMRI study.

**Defining anatomical ROIs.** Anatomical regions were delineated by applying the FreeSurfer anatomical parcellation algorithm[53] to each subject's high-resolution anatomical scan (Fig. 2a). Briefly, this process includes the digital removal of non-brain tissue, automated segmentation of the cerebral cortex, subcortical white matter, brainstem, cerebellum, and deep gray matter volumetric structures (e.g., amygdala, hippocampus, and putamen), generation of a model of each subject's cerebral cortical surface, and automated parcellation of each subject's cortical surface model into anatomical units based on his or her cortical folding patterns. The Desikan–Killiany cortical atlas[32] as implemented in FreeSurfer 5.3[53] was used to assign anatomical labels to each subject's cortical surface model. This gyral-based atlas defines a gyrus as tissue between two adjacent sulci. As such, a particular gyral label in this atlas (e.g., left inferior temporal gyrus) corresponds to both the associated gyrus and the adjacent banks of its limiting sulci. This procedure yielded 34 atlas labels for each hemisphere, as well as 6 labels corresponding to subcortical structures within each hemisphere. Thus, in total, 80 anatomical ROIs were defined for each subject (Supplementary Table 2 and Fig. 3 for a complete list of ROIs).

**Preprocessing of fMRI data.** Preprocessing of fMRI time series data was performed using AFNI[54]. For each run, functional data were despiked using the AFNI program 3dDespike to remove transient, extreme signal fluctuations not attributable to biological phenomena. Next, each subject's functional scans were aligned to his or her anatomical scan using a six-parameter rigid body least squares transformation. Motion parameters from this volume registration step were saved for later removal from the signal time series as regressors of no interest. The first two volumes of each run were discarded in order to avoid including data potentially characterized by large signal changes prior to tissue reaching a steady state of radiofrequency excitation. Each voxel's time series was scaled to its mean within each run.

In addition to motion parameters extracted during volume registration, time series from voxels corresponding to white matter and ventricles were extracted for later inclusion as regressors of no interest, as signal fluctuations in white matter and cerebrospinal fluid largely reflect noise due to subject motion, instrument instabilities, and physiological artifacts, such as cardiac and respiratory effects[55,56]. White matter and ventricle masks were extracted based on each subject's FreeSurfer segmentation file. These masks were eroded to avoid inclusion of gray matter voxels by excluding any voxels with one or more non-white matter neighbors from the white matter mask, and any voxels with two or more non-ventricle voxel neighbors from the ventricle mask. A relatively less conservative erosion threshold was applied to the ventricle masks to ensure that all subjects' ventricle masks contained voxels; these thresholds were chosen based on the recommendations provided by afni_restproc.py. Data were spatially smoothed

separately within gray matter and non-gray matter masks using a 4-mm full width at half maximum Gaussian smoothing kernel. The average time series from each run was extracted from the ventricle mask for use as a global regressor of no interest. In addition, a local regressor of no interest was computed for each voxel by taking the average time series of white matter voxels within a 15-mm radius of that voxel. The temporal derivatives of each regressor of no interest (i.e., motion parameters extracted during volume registration, average ventricle signal, and local white matter signal) were computed for use as additional regressors of no interest. Next, a third-order polynomial was removed from all regressors of no interest to avoid the inclusion of competing polynomial terms during the subsequent regression.

Finally, nuisance signals (i.e., motion parameters, average ventricle signal, local white matter signal, and their derivatives) and a third-order polynomial were regressed out of the preprocessed time series of each voxel for each run for each subject. The goal of this procedure was to remove signal changes dues to subject motion, physiological artifacts (e.g., respiration and cardiac effects), and instrument instabilities in order to provide a better estimate of signal fluctuations due to neural processing. Nuisance variable regression is often employed to attenuate temporal autocorrelation characterizing fMRI response time series, which can bias estimates of error variance and thus, the significance of test statistics describing those time series, due to an underestimation of the true degrees of freedom[57]. In the current study, however, the relative magnitudes of correlation coefficients between corresponding time series (which, unlike corresponding *p*-values would not be biased by temporal autocorrelation within individual time series) were entered into separate statistical analyses investigating how dyadic similarity varied as a function of social distance. Thus, removing the effects of the nuisance variables as described above served primarily to decrease noise in the data unrelated to cognitive and affective processing of the stimuli. For each subject, these preprocessed time series data were concatenated across all six experimental runs. The average preprocessed time series from each of the 80 anatomical ROIs was extracted for each subject (i.e., data were averaged across all voxels within a given ROI at each time point for each subject).

Due to coverage issues, five subjects were missing data for 1 or more ROI. Specifically, two subjects were missing data for a single ROI, one subject was missing data for 2 ROIs, one subject was missing data for 6 ROIs, and one subject was missing data for 21 ROIs. Missing data were concentrated primarily in the temporal lobes (Supplementary Table 2).

**Extracting dyadic similarities of fMRI response time series.** Given that there were 42 subjects in the fMRI component of the study, there were 861 unique (undirected) dyads of fMRI subjects. For each of these 861 dyads, the Pearson correlation between the time series of their fMRI responses was computed for each of 80 anatomical ROIs (Fig. 2). For 1259 of these 68 880 total data points (i.e., 861 subject pairs×80 anatomical ROIs), at least one subject in the dyad lacked data for the corresponding ROI (Supplementary Table 2). In such cases, the correlation value for this dyad was imputed as the average correlation value for that ROI from all remaining dyads. The resulting similarity vectors for each of the 80 anatomical ROIs were normalized to have a mean of zero and a SD of 1 (Fig. 3).

**Predicting social distance based on neural similarities.** As described in the main text, we tested if it would be possible to predict whether two individuals were friends, friends of friends, or farther apart in the social network based on the similarities of their fMRI response time series. If so, it should be possible to build a predictive model of social distance by training an algorithm to recognize patterns

of neural similarities associated with various social distance categories from a subset of dyads' data. This model should then correctly generalize to predicting the social distances characterizing new dyads given data summarizing the similarity of those dyads' fMRI responses to naturalistic stimuli (i.e., from the eighty-element vectors that summarize the similarities of neural responses for each dyad). Given that the current data were imbalanced across social distance categories (i.e., $n = 63$ for distance 1 dyads; $n = 286$ for distance 2 dyads; $n = 412$ for distance 3 dyads; and $n = 100$ for distance 4+ dyads), data resampling and folding procedures were used to create a series of balanced data folds such that all dyads were included in the analyses, as described in more detail below.

First, the data set was divided into eight training and test folds using the StratifiedKFold function in scikit-learn[24], which ensures equivalent percentages of samples of each class across training and test folds. To attenuate problems of class imbalance, sampling techniques such as undersampling (i.e., omitting examples of over-represented classes from the data set) and oversampling (i.e., adding copies of examples from under-represented classes to the data set) are often used. Undersampling can entail excluding a large amount of data from analyses (e.g., in the current study, including only 63 examples of each category would entail using only 252 dyads' data, effectively excluding 609 dyads—71% of the total data set). Oversampling ensures that all examples (here, all dyads' data) are included in analyses. Here oversampling was implemented within each training fold to generate equal numbers of dyads of each social distance category within each training fold. Distance categories containing relatively few dyads within each training fold were made equivalent in size to the larger social distance categories by iteratively sampling randomly without replacement from the examples of the corresponding distance category within that training fold until there was an equivalent number of data points from each category within the training fold. This approach ensures that no data points are entirely excluded from analysis, while ensuring that any overfitting resulting from oversampling will not artificially inflate cross-validated model performance, since oversampling is performed only within each training fold and performance is ultimately assessed within the previously held-out testing data from each fold.

Within the training data of each data fold, a grid search procedure was implemented in scikit-learn[24] to select the hyper-parameter (i.e., the value of the $C$ parameter from a grid of logarithmically spaced values between 0.001 and 1000) of a linear SVM learning algorithm that would best separate items in the training data set according to social distance. More specifically, the training data within each data fold was subdivided into eight additional data folds that were each partitioned into training and validation data sets, and the $C$ value that performed most accurately on validation data across folds within the training data was selected as the best estimator for that data fold. The best estimator was then retrained on all training data from the given data fold, and its out-of-sample predictive performance was tested on the left-out testing data for that data fold. This process was repeated iteratively for each data fold. Results in the main text reflect the average cross-validated predictive performance across data folds.

To compare the actual cross-validated predictive performance to what would be expected based on chance alone, permutation testing was used. The procedure described above was repeated 1000 times while randomly shuffling the labels corresponding to the data in each training fold to estimate a null distribution of cross-validated prediction accuracies corresponding to what would be achieved by random guessing. The distribution and mean of the cross-validated predictive accuracies achieved in the randomly permuted data are illustrated in Fig. 8b.

Data visualization was performed using the python packages PySurfer[58], seaborn,[59] and Matplotlib[60], as well as the R packages igraph[52] and ggplot2[61].

**Code availability**. The code used for the analyses also is available upon request.

**Data availability**. The data that support the findings of this study are available from the corresponding author upon request.

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

## Acknowledgements

This work was supported by a graduate fellowship from the Neukom Institute for Computational Science and a Dartmouth Cross-disciplinary Collaboration Seed Grant.

## Author contributions

C.P., A.M.K., and T.W. designed the study and experiments. C.P. and A.M.K. collected and analyzed the data. C.P., A.M.K., and T.W. wrote the manuscript.

## Additional information

**Competing interests:** The authors declare no competing financial interests.

