## [Peer Review File · Nature Communications]

Reviewers' comments:

Reviewer #1 (Remarks to the Author):

Review of "Neural Homophily: Similar neural responses predict friendship" Nature Communications.

This paper examines an interesting and important question: Are social networking behaviors tied to brain activity patterns? This paper takes a first step at measuring the level of similarity between brain activity and patterns of friendship using a sample of MBA students. The findings indicate that brain activity patterns measured in reaction to video stimuli predict patterns of friendship, suggesting that similar brain activity patterns are associated with similar patterns of friendship. I like the fundamental question the paper is asking and the unique data the authors collected. Nevertheless, I have concerns that I hope the authors can address about the research design. Currently, I think the design cannot ensure valid conclusions were drawn from the analysis. My concerns are:

Social network design

How do you deal with these three types of network sampling bias in your self-reported network data? 1. Sampling from a network creates biases in which links are and are not measured (Granovetter 1976). 2. The sample from the whole MBA class is among students in the same MBA elective course; therefore, those students may have interactions with one another that are not representative of most student interactions. 3. You measure just a piece of the social network of each MBA student. How can you eliminate the possibility that measures of their larger network of friends, colleagues, associates, and work connections beyond their MBA network would not change your results?

It is common to look at all ties in a network, not just reciprocated ties. Why did you only look at reciprocal ties? You may have a justification for looking only at reciprocated ties, but in doing so you are discarding a lot of data. What happens to the results if you look at all ties?

Did you randomize the network to make sure the observed relationships could not be explained by chance? Please include basic statistics on the network.

Regression

The regression analysis adds important control variables, but some variables that I thought you would have included and could include did not seem to be controlled for. For example, there does not seem to be a control for having seen a specific video prior to the study. You control for demographic characteristics of subjects but not the demographic similarity of a pair of subjects and i and j . What happens when you control for those characteristics?

Why isn't there a control for autocorrelation in the regression? Two time series may look to be correlated with each other but the correlation is spurious because the themselves are correlated with themselves. It is common to run several diagnostic tests on TS including autocorrelation and stationarity. Were these tests conducted? What were the results of the

tests, and do they change the presented results?

Were robust SE used in addition to the clustering? If not, why not?

Measures of Neural ROI

For a person unfamiliar with neuroscience practices, it was unclear how similarity in neural ROI was measured. Can the methods be explained in more general terms? Also, didn't you use typical measures of similarity in TSs like the Pearson, Spearman, Kendal, C, or dCor correlation coefficients? If you use the aforementioned measures are the results the same as those reported?

Results

Fig 3. I know you report t stats on the differences between the 1-4 distances, but the error bars appear to tell a different story. According to the fig, there is only a difference between 1 hop and 3 hops. 2, 3, and 4 hops appear not to be different from each other. Am I reading this correctly?

Given that you have lots of dyad measurements, why not use a KS test on the distribution of values rather than a t test, which is subject to distributional assumptions your data might not meet?

Throughout the paper, you talk about NS predicting friendship. It seems the reverse correlation is nearly as strong per the second paragraph of page 6. Can you present a formal Granger test at each hop distance? If the correlation goes in both directions, I still think your findings are interesting and novel. The formal test would tighten the paper.

Also, on page 6, paragraph 1, you say NS "increases" the likelihood. "Increases" suggests a causal relationship, which I know you are proposing. To avoid confusion, I recommend changing the word "increases" to "is associated with."

Framing

The paper is framed overly broadly. For example, the first sentence begins with, "From cells to celestial bodies...". That kind of beginning is best suited for a PopSci book and does little to meaningful frame the study for the reader. It also feels overstated because while your study is novel, it is nonetheless much more down to earth, based on a small convenience sample, no experimental randomization, and nascent theory. I would like to see the paper presented at a more reasonable and meaningful level. As I mentioned above, you have original empirical findings on an important topic that can begin to provide first facts for understanding the link between neuroscience and network science theory.

Reviewer #2 (Remarks to the Author):

In this paper by Parkinson et al., social distance between individuals (1-4 degrees of separation in reported friendships) in a first-year MBA class is predicted by similarity of fMRI

response timecourses during viewing of audiovisual movies. The effect of neural similarity is present after controlling for demographic variables such as age, gender, and nationality. Inter-subject similarity (normalized within ROI and aggregated across the brain) decreased with increasing social distance, up to but not including 4 degrees of separation. In a separate classification analysis, social distance of any given dyad was predicted using the pattern of dyad similarity across ROIs (80 regions spanning the whole brain).

This is a strong paper, demonstrating an interesting social-neuroscientific phenomenon with methodological sophistication. I have a few suggestions for ways in which the paper can be expanded and improved.

1. The main text is around 1700 words and 4 figures, but Nature Communications format is max 5000 words and 10 figures/tables. While I understand that this journal doesn't have a minimum length, the article currently reads as though it's written for a "brief" journal format, with almost no introductory material and a very short discussion. I think the article would benefit greatly from having the background ideas and interpretation of results fleshed out more, and the current format would allow plenty of room for expansion.

2. In their dyad-level regression model, the authors use a weighted average of neural responses across 80 ROIs. I would be very interested to see the model results for each ROI plotted as a brain map (visually like Fig 3A). This map could be highly informative as to whether particular regions or known brain networks are contributing more to the social distance effect; for example, if responses in early visual cortices do an excellent job of predicting social distance, this supports a different interpretation of what's going on as opposed to if responses in theory-of-mind network areas are the most predictive. Such a map analysis would be considered a follow-up to the weighted-average aggregate version already reported, and thus multiple comparisons correction need be applied only subsequently across the 80 ROIs in the map (ie not including the aggregate analysis), if p-values are to be reported at all (depending on the journal demands). Personally I would support the display of betas alone without p-values, betas being an important indicator of effect size and p-values being fairly non-meaningful in this context.

3. The authors note that the dependence structure of the dyadic data may lead to an increase in the risk of Type I error, and cite a few papers detailing the methods used to correct for this bias in the dyad-level regression model. I am not an expert in such methods and have difficulty assessing the efficacy of the correction. An alternative approach would be to remove dyads (randomly when possible) from each social distance category such that each fMRI participant contributes an equal number of times to each category. Hopefully this would be possible while still retaining a reasonable number of dyads per category. I suggest the authors attempt something like this and, if the results are interpretable, provide the results in supplementary material; while more crude than their current approach, I believe it will be useful for convincing readers who are not conversant with the clustering-based bias correction used in the main paper.

4. Were some movies more diagnostic of dyad friendship than others? The authors note in the Methods that because video order was fixed across subjects, it might not be possible "to

identify what kinds of content may have been driving any observed effects". I agree that the fixed order may not present an ideal situation for linking stimulus content to homophily effects, but nonetheless it would be interesting to see the overall model performance, prediction accuracy, or rank for individual movies. After all, a number of other fMRI studies in the literature have succeeded in linking neural and behavioral effects to movie content despite using fixed-order presentation. Even without a detailed analysis of the semantic content of the movies in the current paper, future experimenters may find inspiration in the pattern of results, e.g., by doing their own analyses of the movies' content. For example, these results could be added to Table S1: either a) the model betas, or b) prediction accuracy confusion matrix summarized as the average of cells within each social distance category (4 values).

5. A few more details about the data collection for the MBA social network would be helpful. How far into the year was the survey administered, and is there any information about which, if any, friendships were formed prior to the current year as opposed to during it? How much time passed between the survey and the fMRI data collection -- is there any reason to be concerned that the reported friendships might have changed in the interim? Across how much time were the fMRI data collected? Were the fMRI subjects re-surveyed at the time of the scan?

6. For Figure S2, the authors write that "inter-subject fMRI response time series similarities were normalized within brain region". If I understand the procedure correctly (as described in the Methods), the normalization is performed across the 861 dyads using z-score, within each ROI, and thus we should not be comparing the ROI similarity values to each other -- we should only be comparing between social distance categories. Ie, we should not compare the rows of the table, only the columns of the table. This procedure and display are fine, but I suggest that the authors add some language to the figure legend and Methods to clarify that the ROI similarity values should not be compared to each other; this will make things easier for the reader. I had the automatic response of comparing ROIs (perhaps because column 1 of Fig S2 is sorted), so it took a while to figure out what the correct interpretation was. I believe the same comment applies to Figure 3, and is particularly important for 3A, as the temptation to compare ROI values presented in a map is strong.

7. Fig 3 is described out of order, appearing later than Fig 4 in the text. In fact, Fig 3 is not described until the Discussion (though there are no section headings so I am inferring where the Discussion begins), and it was a bit hard to figure out what analyses were behind 3A and 3B only from the figure legend and cross-referencing with Fig S2 and one paragraph in the Methods. I am unsure how inter-subject similarity scores were aggregated across ROIs in 3B (averaging?) or what the error bars represent. This analysis should probably have its own section in the Results.

8. Fig 4a: Colorbar labels appear to be truncated.

Reviewer #1 (Remarks to the Author):

This paper examines an interesting and important question: Are social networking behaviors tied to brain activity patterns? This paper takes a first step at measuring the level of similarity between brain activity and patterns of friendship using a sample of MBA students. The findings indicate that brain activity patterns measured in reaction to video stimuli predict patterns of friendship, suggesting that similar brain activity patterns are associated with similar patterns of friendship. I like the fundamental question the paper is asking and the unique data the authors collected. Nevertheless, I have concerns that I hope the authors can address about the research design. Currently, I think the design cannot ensure valid conclusions were drawn from the analysis. My concerns are:

Social network design

How do you deal with these three types of network sampling bias in your self-reported network data? 1. Sampling from a network creates biases in which links are and are not measured (Granovetter 1976). 2. The sample from the whole MBA class is among students in the same MBA elective course; therefore, those students may have interactions with one another that are not representative of most student interactions. 3. You measure just a piece of the social network of each MBA student. How can you eliminate the possibility that measures of their larger network of friends, colleagues, associates, and work connections beyond their MBA network would not change your results?

Thank you for your thoughtful comments and suggestions regarding this issue and about our paper in general. Regarding the first point raised here, we note that the Granovetter (1976) paper examines situations in which intractably large networks cannot be studied in their completeness, but must be sampled. In our study, the complete cohort of first-year MBA students was surveyed, so no sampling was needed. The course in which the network data were collected is a required course that all members of the cohort must take. It is not an elective and there is no selection effect into the course (conditional on selection into the program). Every first-year student is included in the study population. We have revised the manuscript to clarify that all members of the cohort completed the social network survey and apologize that the previous wording (“all members of the class”) was ambiguous. Specifically, on p. 19, we now state, *“Participants in Part 1 of the study were 279 (89 females) first-year students in a graduate program at a private university in the United States who participated as part of their coursework on leadership. The total size of the graduate cohort was 279 students (i.e., all students in the cohort participated in the leadership course); a 100% response rate was obtained for Part 1 of the study, which was done in accordance with the standards of the local ethical review board.”* We have also revised the caption for Fig. 1 (p. 33 of the revised manuscript) to more clearly convey this information: *“The social network of an entire cohort of first-year graduate students was reconstructed based on a survey completed by all students in the cohort (N = 279; 100% response rate).”*

We apologize for not clearly communicating this information in the previous version of the paper. We hope that clarifying that all students in the cohort completed the social network survey also allays the second concern raised here (i.e., the extent to which interactions indicated on the social network survey are representative of typical student interactions in the cohort).

Regarding the third point raised here, it is undoubtedly true that the students we study have networks outside their cohort of classmates and that these networks are unobservable to us (e.g., their friends from before they entered the program, prior work colleagues and associates, family, etc.). Because of the small size, intimate culture, and remote location of the university at which the study was conducted, these extra-campus contacts play a relatively small role in students' daily lives compared to their quotidian, face-to-face interactions with their classmates. We see no plausible mechanism by which the existence of such friends outside of the campus would threaten the validity of our results, but agree that it is possible that some social distances between students who are not directly connected to one another may be underestimated due to indirect connections via individuals outside of their graduate cohort. We now summarize these points on p. 20 of the revised manuscript: *“The social network survey used here only inquired about students' interactions with other members of their academic cohort. Participants undoubtedly have interactions with individuals outside of their cohort of classmates that this survey did not measure (e.g., with family members, prior colleagues, friends from before they entered the program, etc.). We note that the current study was conducted at a relatively small and remotely located institution where participants' contacts outside of campus likely play a smaller role in their daily lives compared to their quotidian, face-to-face interactions with their classmates. That said, social distances between some participants who did not report friendships with one another may be underestimated due to indirect connections through individuals outside of the graduate cohort.”*

We would be happy to include any additional information or discussion regarding the possibility of sampling biases in order to fully address any lingering concerns.

It is common to look at all ties in a network, not just reciprocated ties. Why did you only look at reciprocal ties? You may have a justification for looking only at reciprocated ties, but in doing so you are discarding a lot of data. What happens to the results if you look at all ties?

The theoretical rationale for using reciprocal ties is that neural similarity is an inherently undirected dyadic construct, so it makes sense to pair it with an undirected measure of friendship. We planned to look only at mutually reported social ties (rather than unreciprocated social ties) because we reasoned that mutually reported ties comprised a stronger indicator of friendship compared with unreciprocated ties. We now summarize this rationale in the Results section of the revised manuscript (pp. 6-7). That said, our results do fully replicate if we use a graph where edges are defined by all ties (including those that were unreciprocated). These results are summarized on p. 1 of the Supplementary Results section. A portion of the relevant text from p. 1 of the Supplementary Results section is provided here for convenience: *“Our main analyses defined social ties based only on reciprocated ties. We reasoned that some unreciprocated ties may be the result of some participants tending to nominate large numbers of classmates as friends (out-degree ranged from 2 to 146), and that mutually reported ties were most likely to correspond to meaningful friendships. The same pattern of results as is reported in the main text was achieved when defining social distance based on both reciprocated and unreciprocated ties. An ordered logistic regression model revealed a significant effect of neural similarity ($\beta = -0.26$, $SE = 0.12$, $p = .029$) on social distance that was comparable in magnitude*

to our main results: holding other covariates constant, compared to a dyad at the mean level of neural similarity and at any given level of social distance, a dyad one standard deviation more similar is 23% more likely to have social distance that is one unit shorter.”

Did you randomize the network to make sure the observed relationships could not be explained by chance?

As suggested, we have added permutation-based analyses that involve network randomization to the Supplementary Results section of the revised manuscript (pp. 3-4 of the Supplement and Fig. S3). The results of these analyses converge with those reported in the main text, and we think that the revised manuscript has been strengthened by demonstrating that diverse data analytic approaches provide convergent results. For convenience, we include the relevant text below:

***“Permutation testing based on network randomization.** We also performed permutation testing of the data to supplement the analyses described in the main text. We adopted the topological clustering methods employed by Christakis and Fowler (2013) to test if there was a greater degree of clustering of particular neural response patterns than would be expected based on chance (i.e., if there was exceptionally high neural similarity among individuals close together in the social network). This method entailed iteratively computing the neural similarity between all individuals in the network in 1,000 randomly generated datasets in which the topology of the social network and the prevalence of particular neural response patterns were held constant while the assignment of neural data to individuals was randomly shuffled.*

More specifically, a distribution of Pearson correlation coefficients corresponding to the null hypothesis that no relationship exists between social distance and neural similarity was obtained by randomly shuffling the neural time series data among participants 1,000 times, then computing the weighted (by ROI volume, as described in the main text) average neural similarity for dyads in each social distance category for each of the 1,000 randomly generated permutations of the dataset. Each participant’s neural time series data consists of an 80 (brain regions) x 1,010 (time points) matrix – i.e., a set of 80 time series, each consisting of 1,010 time points. These neural time series datasets were randomly shuffled among the 42 fMRI study participants 1,000 times while keeping the social network data characterizing connections between participants constant. The magnitude of the weighted average neural similarity for each social distance category within each of the randomly permuted datasets was compared to that of the original, non-permuted data.

Results of these permutation tests revealed a similar pattern of results to those described in the main text and are illustrated in Fig. S3. Distance 1 dyads’ neural response time series were, on average, exceptionally more similar to one another than would be expected based on chance, $p = .03$. There was a non-significant trend such that distance 2 dyads were marginally more similar to one another than would be expected based on chance, $p = .06$. Distance 3 dyads were exceptionally less similar to one another than would be expected based on chance, $p = .003$. Distance 4 dyads were neither more or less similar to one another than would be expected based on chance, $p = .5$.”

Please include basic statistics on the network.

We apologize for having previously omitted this information. We now describe the network diameter, density, total reciprocity, and dyad-level reciprocity on pp. 6-7 of the Results section in the revised manuscript. In the same section, we also provide individual-level means, medians, and standard deviations for in-degree and out-degree. In addition, in Fig. S1, we now illustrate the in-degree and out-degree distributions for the entire network, as well as for the subset of students who participated in the fMRI study, and we provide the distribution of geodesic distances characterizing all unique dyads in the entire network, as well as in the subset of students who participated in the fMRI study.

Regression

The regression analysis adds important control variables, but some variables that I thought you would have included and could include did not seem to be controlled for. For example, there does not seem to be a control for having seen a specific video prior to the study. You control for demographic characteristics of subjects but not the demographic similarity of a pair of subjects and i and j. What happens when you control for those characteristics?

We have repeated our main analyses excluding any dyads whose members had both seen any of the same videos prior to participating in the study. As reported on pp. 2-3 of the Supplementary Results section, *“The effect of neural similarity on social distance remained significant ($\beta = -0.218$, $SE = 0.107$, $p = .042$) in our main ordered logistic regression analysis if dyads whose members had both seen the same clips before were excluded from analyses.”* We also note that in the analyses of each video clip in isolation reported in Table S3, of the 5 videos for which neural response similarity was significantly associated with social network proximity, 3 had never been seen before by both members of any of the 861 dyads in our sample, one had been seen before by both members of only a single dyad, and only one video had been seen before by both members of multiple dyads (specifically, by 3 of the 861 total dyads). In addition, for the video that had been viewed by the greatest number of participants (‘Life’s Too Short’, which had been viewed previously by 4 participants; i.e., by both members of 6 of our 861 dyads), the relationship between social network proximity and neural similarity did not approach significance ($\beta = 0.22$, $SE = 0.22$, $p = .32$). Unfortunately, excluding any dyads where either or both members had seen any of the video clips before rendered the sample too small to be useful, as it resulted in eliminating nearly one third of the 42 fMRI participants and as a result, over half of the dyads (reducing the number of dyads from 861 to 406). However, given the points summarized above, we are confident that the current pattern of results is not attributable to heightened neural similarity among individuals who had both seen the same stimuli before. We would be happy to carry out any further analyses or provide any additional information regarding this concern.

In addition, we have revised our description of how the control variables were entered into the ordered logistic regression on pp. 8-9 of the revised manuscript in order to clarify that demographic similarity among pairs of subjects was included in the model: *“To account for demographic differences that might impact social network structure, we also included in our model binary predictor variables indicating whether participants in each dyad were of the same or different nationalities, ethnicities and genders, as well as a variable indicating the age*

difference between members of each dyad. In addition, a binary variable was included indicating whether participants were the same or different in terms of handedness, given that this may be related to differences in brain functional organization.”

Why isn't there a control for autocorrelation in the regression? Two time series may look to be correlated with each other but the correlation is spurious because the themselves are correlated with themselves. It is common to run several diagnostic tests on TS including autocorrelation and stationarity. Were these tests conducted? What were the results of the tests, and do they change the presented results?

It is no doubt true that within a given participant's fMRI response time series in a given brain region, response levels corresponding to adjacent time points will be more related to one another than those corresponding to time points further removed from one another in time. When evaluating the statistical significance of the correlation between 2 participants' time series, autocorrelation within each time series could inflate the apparent degrees of freedom (since time points are not independent), thus providing spuriously significant p -values corresponding to that correlation. However, we note that in the current study, we do not provide or interpret the p -values corresponding to correlations between time series. Rather, we enter the correlation coefficients characterizing dyads' neural response time series similarities into separate regression models (in which autocorrelation among observations due to dyadic dependencies in the data are corrected for using cluster-robust estimation of standard errors). We do not report or base inferences on the p -values corresponding to the dyadic correlation coefficients given that such p -values may be under-estimated due to potentially over-estimating degrees of freedom given temporal dependencies among observations from nearby time points.

Rather than interpreting the significance of dyadic correlations themselves, here we compare the magnitude of correlations (without regard to their corresponding p -values or degrees of freedom) across dyads belonging to different social distance categories. Given this, autocorrelation within each subject's time series should not bias our results (i.e., even if Subject A's time series is correlated with itself, that should not render Subject A's time series more significantly correlated with those of his/her friends compared with those of students farther removed from him/her in the social network).

We also note that the standard approaches to preprocessing fMRI response time series used here (described on pp. 25-27 of the revised manuscript) do mitigate temporal autocorrelation somewhat by removing stimulus-independent sources of noise that may induce serial correlations in the data, such as oscillatory noise related to physiological artifacts (e.g., respiration and cardiac pulsation), instrument-related low-frequency drifts, and participant head motion by removing the effects of nuisance signals from time series for each participant during preprocessing (e.g., via polynomial detrending and by partialing out, from each local response time series, variance attributable to nuisance time series, such as time series corresponding to motion parameters, signal fluctuations within ventricles, signal fluctuations in local white matter voxels, and their derivatives).

We agree that this is an important consideration, and now discuss these issues on p. 27 of the revised manuscript: *“Nuisance variable regression is often employed to attenuate temporal*

autocorrelation characterizing fMRI response time series, which can bias estimates of error variance and thus, the significance of test statistics describing those time series, due to an underestimation of the true degrees of freedom. In the current study, however, the relative magnitudes of correlation coefficients between corresponding time series (which, unlike corresponding p-values would not be biased by temporal autocorrelation within individual time series) were entered into separate statistical analyses investigating how dyadic similarity varied as a function of social distance. Thus, removing the effects of the nuisance variables as described above served primarily to decrease noise in the data unrelated to cognitive and affective processing of the stimuli.”

Were robust SE used in addition to the clustering? If not, why not?

Robust standard errors adjust for the (possible) existence of heteroscedasticity in the data. Multi-way clustering accounts for both heteroscedasticity and the autocorrelation structure of the dyadic dataset. Therefore, the approach used here (i.e., robust inference using multi-way clustering; Cameron, Gelbach & Miller, 2011) is robust in the usual sense, in addition to accounting for dyadic dependencies in the dataset. Models that estimate standard errors that are merely robust (but not clustered) will necessarily give standard errors that are smaller (and that create the appearance of greater statistical significance), but that are incorrect, as they do not account for autocorrelation in the data. We have revised the manuscript to clarify that the current approach to estimating standard errors accounts for both possible heteroscedasticity (and thus, is robust in the usual sense) and non-independence of data points: *“Cluster-robust standard errors account for both autocorrelation and possible heteroscedasticity in the data; this method of accounting for dyadic dependence is comparable with approaches such as the quadratic assignment procedure or permutation testing”* (p. 8 of the revised manuscript).

Measures of Neural ROI

For a person unfamiliar with neuroscience practices, it was unclear how similarity in neural ROI was measured. Can the methods be explained in more general terms? Also, didn't you use typical measures of similarity in TSs like the Pearson, Spearman, Kendal, C, or dCor correlation coefficients? If you use the aforementioned measures are the results the same as those reported?

We apologize for not defining the similarity measurement clearly in the previous version of the manuscript. The measure of similarity used was simply the Pearson correlation coefficient between time series of neural responses for each brain region for each dyad. The time series have been preprocessed prior to correlation using standard fMRI data processing procedures that broadly aim to identify and remove artifacts (e.g., due to physiological artifacts, head motion, and instrument instabilities) and increase signal-to-noise ratio. After correlating the preprocessed time series for each brain region for each pair of participants, we have simply normalized the correlation coefficients for each brain region across dyads to have a mean of 0 and a standard deviation of 1 (i.e., z-scored across dyads) prior to visualizing the similarities for each brain region (Fig. 4), data analysis, and aggregation of results across brain regions to create the overall weighted mean neural similarity. We have clarified this procedure on p. 8 of the revised manuscript (which now states, *“Pearson correlations were z-scored across dyads for each ROI*

prior to analysis and visualization in order to characterize the relative degree of synchrony in each dyad relative to other dyads for each brain region.”). We have also clarified this procedure in the revised caption for Fig. 4 (which was Fig. 3 in the previous version of the manuscript). The caption for this figure now stipulates, “In order to illustrate how relative similarities of responses in each brain region varied as a function of social distance, inter-subject time series similarities (i.e., Pearson correlation coefficients between preprocessed fMRI response time series) were normalized (i.e., z-scored across dyads for each region) prior to averaging across dyads for each brain region and overlaying results on an inflated model of the cortical surface for each social distance category. Warmer colors indicate relatively similar responses for a given brain region; cooler colors indicate relatively dissimilar responses for that brain region” (pp. 36-37 of the revised manuscript).

The rationale for z-scoring similarities (i.e., Pearson correlation coefficients) across dyads for each brain region was that we reasoned that brain regions might vary in the extent to which they become coupled across participants overall, as well as in the extent to which that coupling varies across dyads. Thus, we aimed to characterize the similarity of responses in each brain region for each pair of participants relative to similarity of responses in that brain region for all other dyads in the study when aggregating and visualizing the data. To ascertain if our results would have changed if we had not z-scored the Pearson correlation coefficients prior to computing the overall weighted mean neural similarity (aggregated across brain regions), we repeated our primary analysis without z-scoring across dyads for each brain region prior to computing the weighted average neural similarity. We describe these analyses and the corresponding results on pp. 1-2 of the Supplementary Results section. Specifically, we report that: *“Prior to conducting the analyses reported in the main text, correlation coefficients were z-scored for each brain region across dyads in order to have a mean of 0 and a standard deviation of 1. This normalization step was performed to account for the fact that brain regions would likely vary in the extent to which they would become coupled across participants overall, as well as in the extent to which that coupling would vary across dyads, and we sought to characterize how similar neural responses were for a given pair of participants for a given brain region, relative to the similarity of all dyads’ responses for that brain region. We also repeated our main analyses without z-scoring the Pearson correlation coefficients, and found the same pattern of results that is reported in the main text. Specifically, in an ordered logistic regression using social distance as the dependent variable and the dissimilarities in control variables (handedness, ethnicity, nationality, age, gender) and weighted (by ROI volume) average neural similarity (based on the Pearson correlation coefficients between preprocessed time series for each brain region for each unique pair of participants) as predictor variables, there was a significant effect of neural similarity on social distance ($\beta = -0.232$, $SE = 0.108$, $p = .03$) nearly identical in magnitude to the results reported in the main text.”*

The results of the logistic regressions carried out separately for each brain region (now summarized in Fig. 3 and Table 2 of the revised manuscript) are identical if the neural similarity metric included in the model contains the raw Pearson correlation coefficients for neural response time series for each dyad for that region or z-scored (across dyads, within region) correlation coefficients. We would be happy to make any additional modifications to the manuscript or analyses if our responses do not adequately address this point.

Results

Fig 3. I know you report t stats on the differences between the 1-4 distances, but the error bars appear to tell a different story. According to the fig, there is only a difference between 1 hop and 3 hops. 2, 3, and 4 hops appear not to be different from each other. Am I reading this correctly?

Given your suggestion below, we now base our inferences about differences in overall neural similarities between dyads in each social distance category and the average of dyads in the remaining social distance categories on Kolmogorov-Smirnov (K-S) tests, rather than *t*-tests, since these tests make distributional assumptions that our data may not meet. In addition, because K-S tests are sensitive to differences between distributions based on factors other than location (e.g., to differences in spread and shape), we complement these analyses with Wilcoxon rank-sum tests, which are specifically sensitive to differences in location between two distributions. For the same reason, we use Wilcoxon rank-sum tests to conduct pairwise tests between overall neural similarities among dyads in each social distance category. The results of these analyses are reported on pp. 11-12 of the revised manuscript.

We still present the deviation coded point estimates and corresponding 95% CIs for illustrative purposes in Fig. 4d (previously Fig. 3b), and realize that we had not adequately described the data presented in this figure in the previous version of the manuscript. We now provide more details regarding this figure on p. 12 of the revised manuscript: *“In order to illustrate how overall neural similarity varies as a function of social distance while holding all control variables (i.e., handedness, age, gender, ethnicity, nationality) constant, deviation-coded point estimates were computed and are illustrated in Fig. 4d. Deviation coding provides, for each social distance, a point estimate and confidence interval of the difference in neural similarity from the average of the other social distance categories; complete details appear in the Supplement.”*

Our understanding of the way to read the error bars on a graph like the one in Fig. 4d (previously Fig. 3) is to compare one point estimate against another point's confidence interval. This interpretation converges with the results of the complementary non-parametric analyses reported in the main text (on pp. 11-12): Distance 1 dyads were more similar than distance 3 and 4 dyads, but were not significantly more similar than distance 2 dyads ($p = 0.13$, two-tailed), and distance 2 dyads were significantly more similar than distance 3 dyads. Distance 4 dyads, for whom the point estimate was very imprecise, yielding a large confidence interval, were statistically significantly less similar than distance 1 dyads but did not differ significantly from distance 2 or 3 dyads. We hope that our revised description of Fig. 4d, along with the expanded summary of the results of nonparametric tests comparing neural similarity between dyads belonging to different social distance categories, more clearly conveys these results.

Given that you have lots of dyad measurements, why not use a KS test on the distribution of values rather than a t test, which is subject to distributional assumptions your data might not meet?

As suggested, we now perform Kolmogorov-Smirnov (K-S) tests, rather than *t*-tests to compare overall neural similarity among dyads at each social distance level to the remaining dyads in the

sample (please see p. 11 of the revised manuscript), given that, as Reviewer 1 points out, *t*-tests make assumptions about the distribution of our data that may not be well-founded. In addition, given that K-S tests are sensitive to any differences between distributions (i.e., not only in location, but also in spread or shape), and given that we were specifically interested in shifts in location (i.e., central tendencies) between the distribution of similarities corresponding to different social distance levels, we complement the K-S analyses with Wilcoxon rank-sum tests, which are specifically sensitive to changes in location between distributions. These two data analytic approaches provide convergent results, as summarized on pp. 11-12 of the revised manuscript.

Throughout the paper, you talk about NS predicting friendship. It seems the reverse correlation is nearly as strong per the second paragraph of page 6. Can you present a formal Granger test at each hop distance? If the correlation goes in both directions, I still think your findings are interesting and novel. The formal test would tighten the paper.

We agree that it is impossible to ascertain from the current findings whether neural similarity causes friendship or *vice versa*. Given that the current data are cross-sectional and we are unable to obtain additional waves of data from this sample, we are unfortunately unable to perform a Granger causal test with these data. As such, we have revised the wording used to describe the relationship between neural similarity and social network proximity throughout the manuscript in order to avoid implying a causal relationship in either direction. In addition, we now discuss the need for future research to acquire longitudinal data in order to ascertain if neural similarity causes or results from friendship (pp. 17-18 of the revised manuscript): *“Do we become friends with people who respond to the environment similarly, or do we come to respond to the world similarly to our friends? Given its cross-sectional nature, the current study cannot address this question directly. Thus, future longitudinal studies should measure whether inter-subject neural response similarities predict subsequent friendship formation among members of evolving social networks. We anticipate that such studies will find that the exceptional similarity of neural responses among friends reflects both homophily and social influence processes. A large body of research demonstrates that people in our immediate environment influence how we think, feel, and behave, and humans’ embeddedness within social networks causes these social influence effects to reverberate outward in social ties, and thus, to extend beyond those individuals with whom we interact with directly. At the same time, similar people may tend to become connected at higher rates because they find themselves in common situations. Similarly, pre-existing similarities in how individuals tend to perceive, interpret, and respond to their environment can enhance social interactions and increase the probability of developing a friendship via positive affective processes and by increasing the ease and clarity of communication. Future research should extend the current findings by adopting longitudinal experimental designs that afford insight into the extent to which the results observed here reflect homophily, social influence processes or a combination of these phenomena.”*

Also, on page 6, paragraph 1, you say NS “increases” the likelihood. “Increases” suggests a causal relationship, which I know you are proposing. To avoid confusion, I recommend changing the word “increases” to “is associated with.”

Thank you for pointing out that using the word “increases” may suggest a causal relationship that cannot be ascertained from the current dataset. As suggested, we have revised the description of these effects throughout so that where we previously stated that neural similarity “increases” the likelihood of friendship or social network proximity, we now state that neural similarity is “associated with an increased likelihood” of these phenomena. For example, in the results summary in question (p. 6 of the previous version of the manuscript and pp. 9-10 of the revised manuscript), we now state, *“Logistic regressions that combined all non-friends into a single category, regardless of social distance, also yielded similar results, such that neural similarity was associated with a dramatically increased likelihood of friendship, even after accounting for similarities in observed demographic variables. More specifically, a one standard deviation increase in overall neural similarity was associated with a 47% increase in the likelihood of friendship.”*

Framing

The paper is framed overly broadly. For example, the first sentence begins with, “From cells to celestial bodies...”. That kind of beginning is best suited for a PopSci book and does little to meaningful frame the study for the reader. It also feels overstated because while your study is novel, it is nonetheless much more down to earth, based on a small convenience sample, no experimental randomization, and nascent theory. I would like to see the paper presented at a more reasonable and meaningful level. As I mentioned above, you have original empirical findings on an important topic that can begin to provide first facts for understanding the link between neuroscience and network science theory.

Thank you for this suggestion. We have revised the manuscript so that the framing is considerably narrower than the previously submitted version. In particular, the Introduction section has been considerably expanded and revised (pp. 3-6 of the revised manuscript); this section now focuses much more specifically on relevant previous research in order to more meaningfully frame the current work for readers. We agree that the previous framing was overly broad and believe that the changes we have made have substantially improved the manuscript. We appreciate this feedback and would be happy to make any additional changes.

Reviewer #2 (Remarks to the Author):

In this paper by Parkinson et al., social distance between individuals (1-4 degrees of separation in reported friendships) in a first-year MBA class is predicted by similarity of fMRI response timecourses during viewing of audiovisual movies. The effect of neural similarity is present after controlling for demographic variables such as age, gender, and nationality. Inter-subject similarity (normalized within ROI and aggregated across the brain) decreased with increasing social distance, up to but not including 4 degrees of separation. In a separate classification analysis, social distance of any given dyad was predicted using the pattern of dyad similarity across ROIs (80 regions spanning the whole brain).

This is a strong paper, demonstrating an interesting social-neuroscientific phenomenon with methodological sophistication. I have a few suggestions for ways in which the paper can be expanded and improved.

1. The main text is around 1700 words and 4 figures, but Nature Communications format is max 5000 words and 10 figures/tables. While I understand that this journal doesn't have a minimum length, the article currently reads as though it's written for a "brief" journal format, with almost no introductory material and a very short discussion. I think the article would benefit greatly from having the background ideas and interpretation of results fleshed out more, and the current format would allow plenty of room for expansion.

We have revised the manuscript in order to include more thorough Introduction and Discussion sections (which comprise pp. 3-6 and pp. 13-19, respectively, of the revised manuscript), and to include a larger number of display items (i.e., figures and tables). We thank Reviewer 2 for this suggestion, and believe that the manuscript has benefitted significantly from this expansion.

2. In their dyad-level regression model, the authors use a weighted average of neural responses across 80 ROIs. I would be very interested to see the model results for each ROI plotted as a brain map (visually like Fig 3A). This map could be highly informative as to whether particular regions or known brain networks are contributing more to the social distance effect; for example, if responses in early visual cortices do an excellent job of predicting social distance, this supports a different interpretation of what's going on as opposed to if responses in theory-of-mind network areas are the most predictive. Such a map analysis would be considered a follow-up to the weighted-average aggregate version already reported, and thus multiple comparisons correction need be applied only subsequently across the 80 ROIs in the map (ie not including the aggregate analysis), if p-values are to be reported at all (depending on the journal demands). Personally I would support the display of betas alone without p-values, betas being an important indicator of effect size and p-values being fairly non-meaningful in this context.

Thank you for this suggestion. We have completed additional analyses where regressions relating neural similarity to social network proximity are carried out independently for each of the 80 regions of interest, and false discovery rate correction is subsequently applied to the p-values from these 80 analyses. This analysis and the corresponding results are summarized on p.

10 of the revised manuscript and are included below for convenience: *“In which brain regions is neural response similarity associated with social network proximity? To gain insight into what brain regions may be driving the relationship between social distance and overall neural similarity, we performed ordered logistic regression analyses analogous to those described above independently for each of the 80 ROIs. This approach is analogous to common fMRI analysis approaches in which regressions are carried out independently at each voxel in the brain, followed by correction for multiple comparisons across voxels. We employed false discovery rate (FDR) correction to correct for multiple comparisons across brain regions. This analysis indicated that neural similarity was associated with social network proximity in regions of the ventral and dorsal striatum (right nucleus accumbens, right and left caudate, left putamen), the right amygdala, the right superior parietal lobule and left inferior parietal cortex. Regression coefficients for each ROI are shown in Fig. 3, and further details for ROIs that met the significance threshold of $p < .05$, FDR-corrected (two-tailed) are provided in Table 2.”*

As suggested, we also display region-wise betas in what is now Fig. 3 (p. 35 of the revised manuscript). Brain regions where social distance was significantly associated with neural similarity, after accounting for inter-subject similarities in control variables (i.e., demographic variables and handedness) are reported in Table 1. We also discuss the interpretation of the current set of results, in light of the brain regions where neural similarity was significantly associated with social network proximity and the functions associated with these brain regions on pp. 14-15 of the revised discussion section.

3. The authors note that the dependence structure of the dyadic data may lead to an increase in the risk of Type I error, and cite a few papers detailing the methods used to correct for this bias in the dyad-level regression model. I am not an expert in such methods and have difficulty assessing the efficacy of the correction. An alternative approach would be to remove dyads (randomly when possible) from each social distance category such that each fMRI participant contributes an equal number of times to each category. Hopefully this would be possible while still retaining a reasonable number of dyads per category. I suggest the authors attempt something like this and, if the results are interpretable, provide the results in supplementary material; while more crude than their current approach, I believe it will be useful for convincing readers who are not conversant with the clustering-based bias correction used in the main paper.

Thank you for this suggestion. Based on this comment and a similar suggestion from Reviewer 1, we have added permutation-based analyses involving network randomization to the Supplementary Results section of the revised manuscript (pp. 3-4 of the Supplement and Fig. S3). The results of these analyses converge with those reported in the main text, and we think that the revised manuscript has been strengthened by demonstrating that diverse data analytic approaches provide convergent results. For convenience, we include the relevant text below:

“Permutation testing based on network randomization. We also performed permutation testing of the data to supplement the analyses described in the main text. We adopted the topological clustering methods employed by Christakis and Fowler (2013) to test if there was a greater degree of clustering of particular neural response patterns than would be expected based on chance (i.e., if there was exceptionally high neural similarity among individuals close together in

the social network). This method entailed iteratively computing the neural similarity between all individuals in the network in 1,000 randomly generated datasets in which the topology of the social network and the prevalence of particular neural response patterns were held constant while the assignment of neural data to individuals was randomly shuffled.

More specifically, a distribution of Pearson correlation coefficients corresponding to the null hypothesis that no relationship exists between social distance and neural similarity was obtained by randomly shuffling the neural time series data among participants 1,000 times, then computing the weighted (by ROI volume, as described in the main text) average neural similarity for dyads in each social distance category for each of the 1,000 randomly generated permutations of the dataset. Each participant's neural time series data consists of an 80 (brain regions) x 1,010 (time points) matrix – i.e., a set of 80 time series, each consisting of 1,010 time points. These neural time series datasets were randomly shuffled among the 42 fMRI study participants 1,000 times while keeping the social network data characterizing connections between participants constant. The magnitude of the weighted average neural similarity for each social distance category within each of the randomly permuted datasets was compared to that of the original, non-permuted data.

Results of these permutation tests revealed a similar pattern of results to those described in the main text and are illustrated in Fig. S3. Distance 1 dyads' neural response time series were, on average, exceptionally more similar to one another than would be expected based on chance, $p = .03$. There was a non-significant trend such that distance 2 dyads were marginally more similar to one another than would be expected based on chance, $p = .06$. Distance 3 dyads were exceptionally less similar to one another than would be expected based on chance, $p = .003$. Distance 4 dyads were neither more or less similar to one another than would be expected based on chance, $p = .5$."

4. Were some movies more diagnostic of dyad friendship than others? The authors note in the Methods that because video order was fixed across subjects, it might not be possible "to identify what kinds of content may have been driving any observed effects". I agree that the fixed order may not present an ideal situation for linking stimulus content to homophily effects, but nonetheless it would be interesting to see the overall model performance, prediction accuracy, or rank for individual movies. After all, a number of other fMRI studies in the literature have succeeded in linking neural and behavioral effects to movie content despite using fixed-order presentation. Even without a detailed analysis of the semantic content of the movies in the current paper, future experimenters may find inspiration in the pattern of results, e.g., by doing their own analyses of the movies' content. For example, these results could be added to Table S1: either a) the model betas, or b) prediction accuracy confusion matrix summarized as the average of cells within each social distance category (4 values).

As suggested, we have analyzed the data corresponding to each video clip separately and now include results of these analyses in the Supplementary Results (please see Table S3).

5. A few more details about the data collection for the MBA social network would be helpful. How far into the year was the survey administered, and is there any information about which, if any, friendships were formed prior to the current year as opposed to during

it? How much time passed between the survey and the fMRI data collection -- is there any reason to be concerned that the reported friendships might have changed in the interim? Across how much time were the fMRI data collected? Were the fMRI subjects re-surveyed at the time of the scan?

Participants in Part 1 of the study (i.e. social network characterization) were all 279 students in the cohort of a two-year graduate program. The social network data were collected in November of the students' first year of study, which began the preceding August (i.e., social network data were collected after participants had been together on campus for 3-4 months prior to completing the social network questionnaire). Therefore, friendships reported on the questionnaire were formed either during the first 3-4 months of students' first academic year in the graduate program, or prior to entering the program. We unfortunately do not have data on which friendships, if any, were formed prior to participants entering the graduate program. We now summarize this information on p. 19 of the revised manuscript: *"The social network survey was administered during November of students' first academic year in the graduate program, which began the preceding August. Therefore, participants had been on campus together for 3-4 months prior to completing the social network survey, and friendships reported on the survey would have been formed either during participants' first few months on campus or prior to entering the graduate program."*

The social network survey described above was completed by all 279 students in the cohort, and all respondents' data was used to compute social distances between the subset of students ($N = 42$) who completed the fMRI study. The survey was unfortunately not re-administered to the cohort at the time of fMRI data collection. The fMRI study was completed during the last 2 weeks of the following February (i.e., approximately 3 months after the social network data had been collected). We now include this information on p. 22 of the revised manuscript: *"Data collection for the neuroimaging study began mid-way through February during participants' first academic year in the graduate program, and all scanning was completed within two weeks. Therefore, all neuroimaging data was collected approximately three months after the collection of the social network data."*

Thank you for pointing out that we had previously neglected to provide these details regarding data collection. We would be happy to provide any further information.

6. For Figure S2, the authors write that "inter-subject fMRI response time series similarities were normalized within brain region". If I understand the procedure correctly (as described in the Methods), the normalization is performed across the 861 dyads using z-score, within each ROI, and thus we should not be comparing the ROI similarity values to each other -- we should only be comparing between social distance categories. I.e., we should not compare the rows of the table, only the columns of the table. This procedure and display are fine, but I suggest that the authors add some language to the figure legend and Methods to clarify that the ROI similarity values should not be compared to each other; this will make things easier for the reader. I had the automatic response of comparing ROIs (perhaps because column 1 of Fig S2 is sorted), so it took a while to figure out what the correct interpretation was. I believe the same comment applies to Figure 3, and is

particularly important for 3A, as the temptation to compare ROI values presented in a map is strong.

Thank you for bringing this potential source of confusion to our attention. You are correct that because ROI similarity values have been normalized across all dyads within each brain region, comparisons in what was previously Fig. 3 (now Fig. 4 in the revised manuscript) should only be made across levels of social distance within each ROI, rather than across ROIs. As suggested, we have clarified the relevant figure captions. The caption for Fig. 4 (previously Fig. 3; pp. 36-37 of the revised manuscript) now includes the following explanation: *“In order to illustrate how relative similarities of responses in each brain region varied as a function of social distance, inter-subject time series similarities (i.e., Pearson correlation coefficients between preprocessed fMRI response time series) were normalized (i.e., z-scored across dyads for each region) prior to averaging across dyads for each brain region and overlaying results on an inflated model of the cortical surface for each social distance category. Warmer colors indicate relatively similar responses for a given brain region; cooler colors indicate relatively dissimilar responses for that brain region. Please note that because similarities have been normalized across dyads for each brain region, values depicted in this figure should be compared across social distance levels for each brain region, rather than across brain regions within or across social distances.”*

As suggested, we have also added a similar clarification in the caption for Fig. S2 (pp. 7-8 of the revised Supplementary Results section): *“In order to illustrate how relative similarities of responses in each brain region varied as a function of social distance, inter-subject time series similarities (i.e., Pearson correlation coefficients between preprocessed fMRI response time series) were normalized (i.e., z-scored across dyads for each region) prior to averaging across dyads for each brain region within each social distance category. Warmer colors indicate relatively similar responses for a given brain region; cooler colors indicate relatively dissimilar responses for that brain region. Please note that because similarities have been normalized across dyads for each brain region, values depicted in this figure should be compared across social distance levels for each brain region, rather than across brain regions within or across social distances.”*

7. Fig 3 is described out of order, appearing later than Fig 4 in the text. In fact, Fig 3 is not described until the Discussion (though there are no section headings so I am inferring where the Discussion begins), and it was a bit hard to figure out what analyses were behind 3A and 3B only from the figure legend and cross-referencing with Fig S2 and one paragraph in the Methods. I am unsure how inter-subject similarity scores were aggregated across ROIs in 3B (averaging?) or what the error bars represent. This analysis should probably have its own section in the Results.

Thank you for raising these points. To address the first point raised here, we have substantially reorganized the manuscript, and the order in which Figs. 1-4 are mentioned in the main text now corresponds to the numbering of these figures. We have also significantly expanded the Results section in order to provide more details about our analyses. We believe that the clarity of the manuscript has been significantly improved as a result of this reorganization and expansion.

We provide more thorough explanations of the analyses behind Fig. 4 (which was Fig. 3 in the previous version of the manuscript) in both the main text and revised figure caption. To address the specific comment regarding what was previously Fig. 3b (Fig. 4d in the revised manuscript), we have followed your suggestion to use a portion of the Results section in the main text to explain how this figure was produced (please see p. 12 of the revised manuscript). Specifically, we now state, *“In order to illustrate how overall neural similarity varies as a function of social distance while holding all control variables (i.e., handedness, age, gender, ethnicity, nationality) constant, deviation-coded point estimates were computed and are illustrated in Fig. 4d. Deviation coding provides, for each social distance, a point estimate and confidence interval of the difference in neural similarity from the average of the other social distance categories; complete details appear in the Supplement.”*

The details of the deviation procedure provided on p. 3 of the Supplement are provided below for convenience:

“Deviation coding of estimates in Figure 4d. *There are many ways to code categorical variables for regression. Conventional dummy coding (where each observation gets a value of 1 for its category and a 0 for other categories) is useful for comparing all other categories against a single “baseline” category. Deviation coding is more appropriate for comparing each category against the overall mean of the sample. In this case, deviation coding measures, for each social distance, a point estimate and confidence interval of the difference in neural similarity from the average of the other social distance categories, after partialing out the effects of control variables (age, nationality, ethnicity, gender, and handedness). To make these estimates, we first calculated deviation-coded dummy variables corresponding to each value of social distance, 2 through 4. Unlike conventional coding of dummy variables, deviation-coded dummy variables take a value of -1 when social distance is equal to one. These deviation-coded dummy variables are then entered, together with variables describing inter-subject differences in demographic variables and handedness, into an ordinary least squares regression model of the standardized, weighted neural similarity measure. As in our primary analyses, estimates were clustered simultaneously on both members of each dyad. The point estimate and confidence interval for distance one dyads were estimated from the intercept; point estimates and confidence intervals for dyads at distances two through four were estimated from their respective deviation-coded variables.”*

We also clarify the meaning of the point estimates and error bars in Fig. 4d (which was previously Fig. 3b) in the figure caption on p. 37 of the revised manuscript: *“Deviation-coded point estimates and 95% CIs for weighted average neural similarities, after accounting for inter-subject similarities in control variables (demographic variables and handedness). Deviation coding measures the difference in overall neural similarity between dyads within each social distance category and the average overall neural similarity of dyads in the other social distance categories, after removing the effects of control variables. For further details on deviation coding, please refer to the Supplement.”*

We would be happy to provide any further analysis details or to reorganize the revised manuscript in order to further improve the clarity with which we present our methods and results.

8. Fig 4a: Colorbar labels appear to be truncated.

Thank you for bringing this attention. We have revised Fig. 5a (which was previously Fig. 4a) to remedy this issue (please see p. 38 of the revised manuscript).

References

- Cameron, A. C., Gelbach, J. B. & Miller, D. L. Robust inference with multiway clustering. *J. Bus. Econ. Stat.* **29**, 238–249 (2011).
- Christakis, N. A. & Fowler, J. H. Social contagion theory: Examining dynamic social networks and human behavior. *Stat. Med.* **32**, 556–77 (2013).

[Editorial Note: Between the first and second round of reviews, the editor mediated a discussion between the authors and Reviewer 1 to clarify some of the latter's concerns.]

REVIEWERS' COMMENTS:

Reviewer #1 (Remarks to the Author):

I had only one major issue with the paper. After learning more about what the authors' constraints are in being able to respond to my comments, I now feel satisfied with their response and that the paper is suitable for publication in Nature Communications.

Reviewer #2 (Remarks to the Author):

This manuscript has been extensively revised, including expansion of the background/discussion, results (figures and tables), and methodological details. I especially appreciate the new Figure 3 and the addition of the permutation test (Figure S3). The authors have done an excellent job of addressing each comment and question, and I feel that the changes made in response to both reviewers have greatly strengthened the paper.

One more comment: the permutation test highlights the finding that dyads at a social distance of 3 are not merely less similar to each other than those at distances of 1 and 2; in fact they are significantly less similar to each other than would be expected by chance. It's unclear to me whether I *should* be trying to interpret this dissimilarity. In several other popular fMRI similarity measures, commonly-used normalization steps can lead to negative correlation values that *should not* be interpreted as "anti-correlations"; e.g., in functional connectivity, if global signal regression has been applied, it is only valid to assess the relative magnitudes of correlations between pairs of brain regions, not the raw magnitudes (e.g., Weissenbacher et al. 2009 NeuroImage). An analogous situation may be present in the current paper; I am not totally sure. I do not believe this issue impacts the main points or soundness of the paper in any way, I merely raise this as a matter of interest for the authors. It will likely enter some readers' minds and thus might be worth a brief mention in the Discussion.

Point-by-Point Response to Reviewers

Reviewer #1 (Remarks to the Author):

I had only one major issue with the paper. After learning more about what the authors' constraints are in being able to respond to my comments, I now feel satisfied with their response and that the paper is suitable for publication in Nature Communications.

We thank Reviewer 1 for thoughtful comments and suggestions throughout the review process.

Reviewer #2 (Remarks to the Author):

This manuscript has been extensively revised, including expansion of the background/discussion, results (figures and tables), and methodological details. I especially appreciate the new Figure 3 and the addition of the permutation test (Figure S3). The authors have done an excellent job of addressing each comment and question, and I feel that the changes made in response to both reviewers have greatly strengthened the paper.

One more comment: the permutation test highlights the finding that dyads at a social distance of 3 are not merely less similar to each other than those at distances of 1 and 2; in fact they are significantly less similar to each other than would be expected by chance. It's unclear to me whether I *should* be trying to interpret this dissimilarity. In several other popular fMRI similarity measures, commonly-used normalization steps can lead to negative correlation values that *should not* be interpreted as "anti-correlations"; e.g., in functional connectivity, if global signal regression has been applied, it is only valid to assess the relative magnitudes of correlations between pairs of brain regions, not the raw magnitudes (e.g., Weissenbacher et al. 2009 NeuroImage). An analogous situation may be present in the current paper; I am not totally sure. I do not believe this issue impacts the main points or soundness of the paper in any way, I merely raise this as a matter of interest for the authors. It will likely enter some readers' minds and thus might be worth a brief mention in the Discussion.

We thank Reviewer 2 for insightful comments in response to both the original and latest version of this manuscript. Reviewer 2 is correct that the finding that distance 3 dyads are significantly less similar to one another than would be expected based on chance does not imply that members of distance 3 dyads have anti-correlated neural response time series. Rather, distance 3 dyads merely have relatively less similar neural response time series than would be expected based on chance. Distance 3 dyad members' neural response time series are still positively correlated with one another, consistent with a large body of previous work showing significant coupling of fMRI response time series while watching video stimuli, in which participants are often strangers. Magnitudes of fMRI response similarities are merely lower among actual distance 3 dyads than among the fictive distance 3 dyads that result from randomly shuffling fMRI responses across nodes in the network while keeping the topological structure of the network constant. To clarify this point, we have added the following language to the Supplementary Note 5 section (p. 12 of the revised Supplementary Information document): *"We note that that the fact that distance 3*

dyads were significantly less similar to one another than would be expected based on chance alone does not imply that members of these dyads had anti-correlated neural response time series. Rather, members of distance 3 dyads were characterized by neural response similarities that were smaller in magnitude than would be expected if there were no relationship between neural response similarity and proximity in the social network.”